



# Analysing CMIP5 EURO-CORDEX models in their ability to produce south foehn and the resulting climate change impact on frequency and spatial extent over western Austria

Philipp Maier[1], Fabian Lehner[1], Tatiana Klisho[1], and Herbert Formayer[1]

[1]Institute of Meteorology and Climatology, University of Natural Resources and Life Sciences, Vienna, Austria

**Correspondence:** Philipp Maier (philipp.maier@boku.ac.at)

**Abstract.** Foehn has an impact on various climatological variables like temperature and humidity in the highly populated valleys of western Austria. With increasing global warming, the question arises as to how well climate projections are able to produce conditions for foehn and how their occurrence changes with climate change. This study uses six XGBoost models to classify south foehn in EURO-CORDEX climate models of CMIP5 generation for two spatial extents (localised and widespread) and three regions Vorarlberg, Tiroler Oberland and Tiroler Unterland in western Austria, located in the Eastern Alps. For each region, a model for distinguishing foehn from no foehn and one to distinguish the foehn event's spatial extent is trained. Several meteorological inputs on pressure levels from ERA5 reanalysis in combination with training data derived from semi-automated weather station data with Objective Foehn Classification are used in the training process. Weights for individual models are derived by analysing the performance of EURO-CORDEX models in their ability to produce south foehn and considering their independence from each other. The performance of individual EURO-CORDEX models is hereby evaluated by analysing their biases for annual occurrence, seasonal accuracy and inter-annual variability in comparison to the training data.The training data confirm other studies by showing that the three selected regions behave differently in their south foehn occurrence and in the portion of widespread events. Bias analysis shows a pronounced negative bias in annual foehn occurrence for models driven by the general circulation model ICHEC-EC-EARTH or MOHC-HadGEM2-ES. EURO-CORDEX models perform similar in capturing south foehn's seasonality, but highly vary in reproducing the inter-annual variability in the historical period. A weighted trend analysis for future behaviour of south foehn in the 21[st] century shows a slight decrease in south foehn frequency under increasing warming conditions in the Tirol regions but an increase in widespread events in all regions, most pronounced in Vorarlberg at the strongest warming. Further, a shift in foehn seasonality can be observed in all regions with a higher frequency in the spring months and a lower frequency from July to October, also depending on the climate change signal.





## 1 Introduction

According to WMO (1992), foehn is a "wind warmed and dried by descent, in general on the lee side of a mountain". Although a localised phenomenon, foehn has an impact on various (eco-)systems due to the corresponding change in meteorological

parameters like temperature and wind speed, ranging from enhancing the melt of Greenland ice sheets (Mattingly et al., 2023) and the Antarctic Larsen C ice shelf (Elvidge et al., 2020) to damaging rise crops (Kusaka et al., 2021), favouring vine growth (Sfîcǎ et al., 2014) and modifying ozone transport in the Alps (Weber and Prévôt, 2002; Baumann, 2001; Seibert et al., 2000; Richner and Hächler, 2013). As foehn, which is driven by distinguished synoptic conditions (Richner and Hächler, 2013), replaces valley air masses by its dry-adiabatic descent into valleys, meteorological parameters like temperature and

humidity change drastically (Steinacker, 2006). Vegetation responds strongly to foehn as the potential evapotranspiration is dependent on air temperature, wind speed, humidity and radiation (FAO, 1998), which are all influenced by foehn. Therefore, it is relevant to study the change of foehn occurrence and its spatial extent in the wake of climate change to better understand the aforementioned impacts implied by foehn. Western Austria hereby is of special interest due to the densely populated Inn and Rhine valleys, a dense network of stations and an economically strong forestry sector, which could be affected by severe

foehn storms (Stucki et al., 2015).

Foehn research has a rich history and was initiated in the Alpine regions by Hann (1866) and carried out in several international field experiments (Seibert, 1990; Mayr et al., 2004). Especially the Wipp Valley south of the city of Innsbruck has a special role for foehn research as it allows optimal foehn conditions by being perpendicular to the Alps' crest and having a steep slope (Zängl et al., 2004; Gohm et al., 2004; Zängl and Gohm, 2006). Gohm and Mayr (2004) even utilised the Wipp Valley

to display the hydraulic foehn theory, describing the descent of foehn into the valleys due to critical flow conditions. Over the history of foehn science, forecasting foehn at station locations has advanced from subjective expert's knowledge to Objective Foehn Classification (OFC) introduced by Vergeiner (2004) and Drechsel and Mayr (2008), where the potential temperature difference between a crest and a valley station is used to classify foehn. Plavcan et al. (2014) used a statistical mixture model to further improve OFC. However, to investigate foehn behaviour in the future, the analysis method has to be adjusted to also

work with the output of climate projections and the absence of station data.

Current reanalysis products and regional climate projections in Europe provide a spatial resolution of up to 9 km (Muñoz-Sabater et al., 2021; Jacob et al., 2014). While this is the minimum for providing reasonable backward trajectory analysis of warming events caused by foehn (Ishizaki and Takayabu, 2009), a resolution of 9 km can not realistically depict the valleys of western Austria as shown in the red box of Fig. 1a) on a resolution of 30 arcsec obtained from Karger et al. (2017). Foehn events

can therefore not be localised correctly and traditional foehn classification methods cannot be applied. However, Sprenger et al. (2017) and Mony et al. (2021) showed that machine learning techniques are capable of connecting coarse meso-scale data from numerical weather predictions (NWPs) with localised foehn occurrences in valley stations. This opens the possibility to analyse climate projections in their ability to produce conditions for foehn.

In this study, a subset of climate projections from the World Climate Research Program Coordinated Regional Downscaling

Experiment (EURO-CORDEX, Jacob et al., 2014) of the fifth phase of the Coupled Model Intercomparison Project (CMIP5,



Taylor et al., 2012) is analysed in their ability to produce foehn-enabling conditions by comparing it to historical station observations and the state-of-the-art European Centre for Medium-Range Weather Forecasts (ECMWF) reanalysis data of the fifth generation (ERA5, Hersbach et al.,2020). Their performance is afterwards used to weight individual models within the climate projection ensemble. Within this weighting process, usually a considerable degree of subjectivity is present due
to the choice of the specific metrics to validate the performance and their combination into weights (Gleckler et al., 2008; Christensen et al., 2010; Knutti et al., 2010; Weigel et al., 2010; Knutti et al., 2017). Different approaches exist, reaching from comparing model probability density functions of e.g. temperature or precipitation to those of observations (Kjellström et al., 2010) to combining model performance compared to observations with an independence metric of individual projections to each other (Knutti et al., 2017; Brunner et al., 2019). In all approaches the underlying assumptions are that uncertainties
and projection spread will be reduced by attributing models, which perform better for a specific case in the historical period, a greater weight (Christensen et al., 2010). Disadvantages of ensemble weighting are, that not all model qualities can be captured in the performance metric due to the large number of degrees of freedom in climate models (Christensen et al., 2010), risking that the models performing best in the historical period may not be best in reproducing climate change signals (Sperna Weiland et al., 2021; Knutti et al., 2017) and the assumption of stationarity under a changing climate (Christensen
et al., 2008; Buser et al., 2009). However, for cases where obvious model performance criteria exist, model weighting is likely to improve predicting a model mean and smaller uncertainties, whereas the choice of the metric is critical (Christensen et al., 2010; Knutti et al., 2017). Given its binary nature, foehn occurrence serves as suitable performance metric by directly reflecting the impact, which is subject of the study. By weighting the EURO-CORDEX models according to their performance, a trend for the future behaviour of south foehn occurrence and its spatial extent over western Austria dependent on global warming is
derived.

For that purpose, outline of the article is as follows. In section 2, the data used in this study are described (section 2.1), the derivation of the training data from station data using OFC is explained (section 2.2), a brief overview of the used machine learning techniques is given (section 2.3), and the method of how the performance of the EURO-CORDEX models is ranked in regard to producing south foehn is stated (section 2.4). The results, further divided into a subsection for the training data
(section 3.1), the training process (section 3.2), the biases and weights for the individual models (section 3.3) and a derived trend for future occurrence and spatial extent of south foehn (section 3.4) are shown in section 3. Section 4 concludes and interprets the main results and gives an outlook on further research in this area.



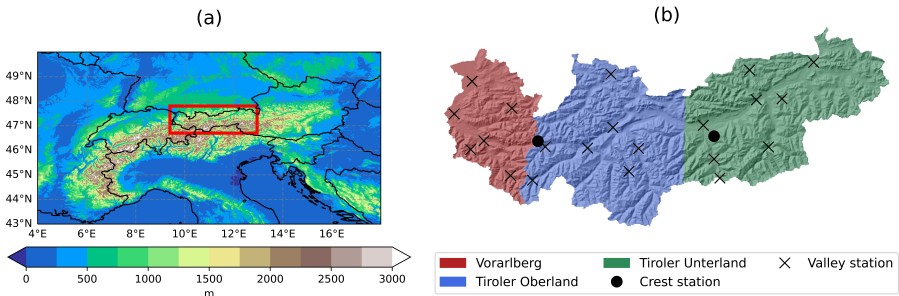

**Figure 1.** (a) Topography of the Alpine area at a resolution of 30 arcsec in metres (Karger et al., 2017). The study area is boxed in red. (b) Coloured subdivision of the study area into the three regions Vorarlberg, Tiroler Unterland and Tiroler Oberland. The locations of the stations used for creating the training data are marked and described in appendix A. The hillshades in the background are adapted from a 10 m digital elevation model (Land Kärnten, 2023).

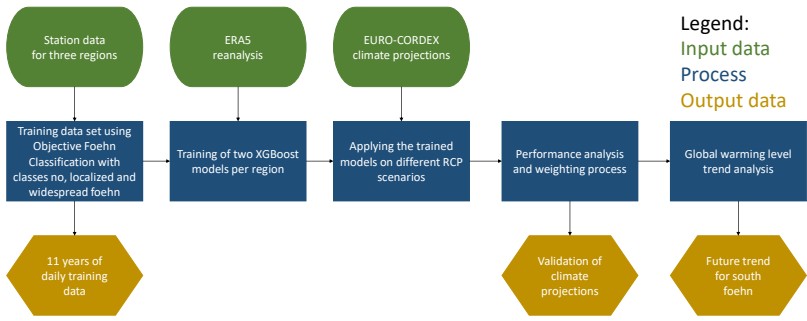

**Figure 2.** Flowchart of the procedure used to derive a future trend for south foehn over western Austria stating the used input data in green, methodological steps in blue and output in yellow. Arrows indicate the process direction.

## 2 Methods

Figure 2 depicts a flowchart of the applied method. Observations, a state-of-the-art reanalysis, and the XGBoost machine learning technique are used to detect south foehn over western Austria for the three Austrian regions Vorarlberg, Tiroler Unterland and Tiroler Oberland. One XGBoost model per region is utilised to detect foehn from no foehn events and another to separate detected events into two different spatial extents (localised and widespread). The trained machine learning technique is then applied on a selection of EURO-CORDEX climate projections and their performance is analysed. A trend for south foehn occurrence and its spatial extent depending on the Global Warming Level (GWL) over western Austria is derived, which is done by weighting the climate projections according to their performance and independence.



## 2.1 Data

ERA5 is the current generation of ECMWF's reanalysis (Hersbach et al., 2020). This data set provides hourly data with a spatial resolution of 0.25°, which roughly translates to 30 km in the European latitudes for the surface and several pressure levels. ERA5 was generated by adjusting a NWP to observations including among others radars, satellites and radiosondes (Hersbach et al., 2020). Variables which are involved in the physical process of foehn formation are used as predictors in the feature matrix (Drechsel and Mayr, 2008). The selection of variables is limited by their availability in EURO-CORDEX models to ensure the applicability of the method to climate projections. Therefore, temperature and specific humidity converted to relative humidity, both at 850 hPa, wind components and geopotential at 500 hPa and mean sea level pressure are used to generate the feature matrix for the machine learning model. Richner and Hächler (2013) state that cross-alpine pressure gradients are the strongest drivers for foehn, therefore the last variable is expected to have particular relevance. The study's domain encompasses the Alpine area, reaching from 44°-50° latitude and 8°-17° longitude, resulting in $37 \times 25$ ERA5 grid cells. As most EURO-CORDEX models only provide daily data, training is done with the daily mean values of ERA5 data. Also the cosine of the day of the year is included to capture seasonality. The variables used in the feature matrices are described in more detail in Table 1.

EURO-CORDEX is an ensemble of Regional Climate Models (RCMs, Jacob et al., 2014) with 12.5 km horizontal resolution, created by dynamically downscaling general circulation models (GCMs) to better represent climate change dynamics. The same variables as in ERA5 are used for three Representative Concentration Pathways (RCPs), RCP 2.6, RCP 4.5 and RCP 8.5 on a daily basis (Van Vuuren et al., 2011). Chimani et al. (2016) state that a subset of EURO-CORDEX models called OEKS15 is suitable for covering a variety of possible climate scenarios in the Austrian domain, while others underperform in this region. Therefore, the ensemble is limited to the OEKS15 selection. As the resolution is different from ERA5, with which the training is done, the models are patch-regridded onto the coarser ERA5 grid (Jiawei Zhuang et al., 2023). Grid-wise bias adjustments on the climate projections are not performed due the danger of altering spatial patterns (Lehner et al., 2023). To ensure that systematic biases are small, all variables are used as differences between two points except the wind speed in 500 hPa, which absolute values were expected to be captured realistically due to conditions close to the free atmosphere (Brands et al., 2011, 2013; Van Nieuwenhuyse et al., 2023). In this study, the names of the individual models are abbreviated. An explanation of abbreviations can be found in appendix C.

For generating the training data, publicly available hourly station observations are used, provided by GeoSphere Austria (2020). To be able to apply OFC, pressure and temperature data to calculate potential temperature and wind information at station locations are used. The stations are selected based on their vicinity to known foehn slopes or to typical outlets of foehn air masses and data availability within the chosen training period 2011-2021, which is the longest period of uninterrupted data for all stations. The station locations are visible in Fig. 1b) and a list of their names, coordinates and elevation can be found in appendix A.



**Table 1.** Feature matrix input variable summary.

| Variable | Pressure level [hPa] | Description | Number of data points |
|---|---|---|---|
| $doy$ [-1...1] | - | Cosine of day of year multiplied by $\frac{2\pi}{365}$ (366 for leap years) | 1 |
| $\Delta p$ [Pa] | sfc | Cross-alpine pressure gradient $\pm 1.0°$ latitude from model main ridge | 37 (length of model main ridge) |
| $\Delta\theta 850$ [K] | 850,sfc[†] | Cross-alpine potential temperature gradient $\pm 1.0°$ latitude from model main ridge | 37 (length of model main ridge) |
| $ws500$ [ms$^{-1}$] | 500 | Total wind speed | 925 (every grid cell) |
| $wd500$ [0...1] | 500 | Wind direction in fraction of north wind speed component to total | 925 (every grid cell) |
| $\Delta Z500$ [m$^2$s$^{-2}$] | 500 | Horizontal geopotential difference to model main ridge on same longitude | 925 (every grid cell) |
| $\Delta rh850$ [%] | 850 | Relative humidity difference to monthly mean | 796[*] (every grid cell above 1200 m) |

While the cross-alpine pressure and potential temperature gradient are used once for every longitude point in the domain, the other variables were used on every valid grid cell. [†]For potential temperature, values at 850 hPa are used as long they are located above the topography, otherwise, the surface (sfc) values are used. [*] Relative humidity is only used where the elevation is below 1200 m to ensure that the 850 hPa surface is above the topography in all EURO-CORDEX models.

## 2.2 Training Data

As the valleys of western Austria like the Wipp and Rhine Valley already served as a location for numerous aforementioned
125 studies for foehn, including studies for developing and testing OFC (Vergeiner, 2004; Drechsel and Mayr, 2008), western
Austria is considered suitable for evaluating the machine learning approach suggested by Mony et al. (2021) and is selected
as study area. To create the training data set, the study area is separated into different regions. Zängl et al. (2004) state that
foehn flow characteristics in the east and west of the study region differ significantly due to the fact that the Rhine valley in
Vorarlberg ends in the Alpine foreland while the Wipp Valley in Tiroler Oberland ends in the east-west running Inn valley. In
130 their study, this led to different timings in foehn breakdown and different flow directions. Therefore, Vorarlberg and Tirol are
treated as separate regions. Moreover, the optimal flow conditions for the Wipp Valley imply that more frequent foehn events
will be observed in the vicinity compared to the rest of Tirol. Therefore, stations in the Wipp Valley and the easterly outflow
of the Inn valley are grouped together in the Tiroler Unterland, in contrast to the Tiroler Oberland, where widespread foehn
events are expected with less frequency. The three regions are visible in Fig. 1b).
135 The years 2011-2021 were determined to be the longest period, where all selected semi-automatic stations delivered consistent data. Therefore, these 11 years are chosen as training period. OFC derived by Vergeiner (2004) and Drechsel and Mayr



(2008) is selected to classify foehn events as binary classes are required and a statistical mixture model yielding foehn probabilities is not suitable for this purpose. However, the thresholds on hourly values for OFC are derived from the application of the statistical mixture model by Plavcan et al. (2014), as shown by Fig. 4a) in the article. Values yielding a foehn probability of 99.9 % in this graph are selected as hard OFC thresholds, visible in Table 2. Valley stations staying within respectively exceeding the thresholds are classified as affected by foehn. As the Tirol areas provide a denser network, at least two stations have meet the OFC thresholds, whereas in Vorarlberg one station suffices for an event to be registered. To consider two different spatial extents of foehn, we define two foehn types, namely localised and widespread foehn. For localised foehn, less than half of the available valley stations meet the OFC thresholds. For a widespread event, more than half the stations (three for Vorarlberg, four for Tiroler Oberland and Unterland) are required to meet the OFC thresholds. Also, different thresholds for the crest wind speed for distinguishing between localised and widespread foehn are applied as higher wind speeds at the crest typically increase the probability for foehn to break into the valleys (Plavcan et al., 2014). The use of percentiles for the crest wind speed is required because the two selected crest stations, Valluga for Vorarlberg and Patscherkofel for the Tirol areas, are prone to different wind characteristics. While the $50^{\text{th}}$ percentile for both stations is similar (4.6 ms$^{-1}$ for Valluga respectively 5.4 ms$^{-1}$ for Patscherkofel in the training period), the $90^{\text{th}}$ percentile differs significantly (8.4 ms$^{-1}$ respectively 14.8 ms$^{-1}$) due to Patscherkofel's gap flow characteristics (Mayr et al., 2004; Gohm and Mayr, 2006). After applying the thresholds on hourly values, the results are aggregated on a daily basis, whereas the most extended event observed within the day's 24 hours is used. The training data (Maier, 2024) therefore comprises 11 years of daily information of the classes 'no foehn', 'localised foehn', and 'widespread foehn' for the three regions:

$$
y_i = \begin{cases} 1 & \text{if foehn is widespread,} \\ 0.5 & \text{if foehn is localised,} \\ 0 & \text{if foehn is not observed,} \end{cases} \tag{1}
$$

where $y_i$ are the training labels used in the machine learning model.

**2.3 XGBoost training**

To be able to identify foehn independent from station data in the climate projections, the supervised machine learning technique XGBoost (Chen and Guestrin, 2016), which is a scaleable tree boosting system, is applied. This study provides detailed information about the training process, while only the formal goal of such an algorithm is stated here. The suitability of XGBoost for different applications in earth sciences has been shown in several studies (Huang et al., 2021; Dong et al., 2023; Razavi-Termeh et al., 2023). Mony et al. (2021) show that XGBoost is suitable in connecting spatial meso-scale weather patterns with foehn observed at two stations in Switzerland. Given its proximity to the study region, a similar approach using the same machine learning model is used.

The model gives predictions $\hat{y}_i$ using a feature matrix $\mathbf{X}$, consisting of $k$ features $\mathbf{x}_i \in \mathbb{R}^k$ from $j$ events and their corresponding true labels $y_i$. XGBoost is used for the classification of two classes (foehn or no foehn respectively localised or





**Table 2.** OFC thresholds applied on hourly station data from the period 2011-2021.

| Threshold | Value |
|---|---|
| Crest wind speed percentile for localised events | 50[th] |
| Crest wind speed percentile for widespread events | 90[th] |
| Valley wind speed [ms$^{-1}$] | 3 |
| Potential temperature difference (crest minus valley) [K] | [-1,3] |
| Wind direction on the crest [° S] | ±45 |
| Number of stations responding for localised event | Vorarlberg: 1-2 |
|  | Tiroler Oberland: 2-3 |
|  | Tiroler Unterland: 2-3 |
| Number of stations responding for widespread event | Vorarlberg: 3-6 |
|  | Tiroler Oberland: 4-7 |
|  | Tiroler Unterland: 4-8 |

widespread foehn), therefore $y_i \in \{0,1\}$. A convex loss function $L_i$ for the $i$-th event is used to validate each training step, aiming for minimisation during process. The two loss functions used are the error, which is defined as the number of wrongly attributed events divided by the number of all events $j$:

$$L_i(y_i, \hat{y}_i) = \frac{\mathrm{abs}(y_i - \hat{y}_i)}{j}, \qquad (2)$$

and the logistic loss, which is defined by

$$L_i(\hat{y}_i, p) = -(\hat{y}_i \log(p) + (1 - \hat{y}_i) \log(1 - p)), \qquad (3)$$

where $p = \Pr(y = 1)$ is the probability for a label $y_i$ to be attributed to the positive class (Janocha and Czarnecki, 2017). Additionally to the loss function, a term punishing model complexity is added to prevent over-fitting to the training (Chen and Guestrin, 2016). The training process is done by modifying the model weights so the combined loss function and complexity term are minimised. If training is successful, a properly minimised loss function results in a high score in accuracy

$$\mathrm{acc} = \frac{\mathrm{TP} + \mathrm{TN}}{\mathrm{TP} + \mathrm{TN} + \mathrm{FA} + \mathrm{ME}}, \qquad (4)$$

which is the sum of true positive TP and true negative TN predictions divided by the sum of TP, TN, the false alarms FA and the missed events ME. For imbalanced data sets, the acc score puts too much emphasis on the dominating class as even a



model never predicting the positive class (resulting in $\mathrm{TP} = \mathrm{FA} = 0$ and maximal $\mathrm{TN}$) would yield high scores of $\mathrm{acc}$, as $\mathrm{ME}$ is small (Murphy, 1996; Hoens and Chawla, 2013). Therefore, it is not suitable to be the only parameter to be evaluated and all the individual metrics resulting in $\mathrm{acc}$ will be observed.

The 11 year long daily training data is used together with the ERA5 variables visible in Table 1. This results in a feature matrix of $j \times k = 4018$ days $\times 3646$ variables. For every region, one model for distinguishing foehn from no foehn and one to

define the foehn event's spatial extent to either localised or widespread is trained with the labels $y_i$ obtained from the training data. This leads to six XGBoost models in total. For optimal training, early stopping and balanced class weightings are used and a hyperparameter grid search using cross validation of five random sets of the feature matrix is performed. The hyperparameters yielding the best accuracy for every model during the grid search can be found in appendix B.

## 2.4  Climate projection performance ranking and weighting

After training the six XGBoost models, they are applied to all $M = 48$ selected EURO-CORDEX models for the period 1991-2100 by creating similar feature matrices for every model. As this study is focused on foehn occurrence and its behaviour, it is intuitive to select the foehn events occurring in the historical period as performance metric. We define three metrics as valuable for a model to capture foehn occurrence accurately, namely (a) how often foehn occurs annually, (b) when it occurs within a year, capturing its seasonality and (c) how individual years differ from each other, indicating the inter-annual variability of

weather. Therefore, to be able to assess the performance of the individual climate projections in producing foehn events $E$, biases $b$ of three different types within the historical period (1991-2020 with $N = 30$ years) of every model, denoted with a subscript $m$, are compared to the training period (2011-2021 with $n = 11$ years), denoted with a subscript $t$. The following biases are calculated:

– The annual-event bias $b_{m,\mathrm{annual}}$ is calculated as the difference of the average yearly number of events $E$:

$$b_{m,\mathrm{annual}} = \frac{1}{N}\sum^{N} E_{m,\mathrm{yearly}} - \frac{1}{n}\sum^{n} E_{t,\mathrm{yearly}} \,. \tag{5}$$

– The seasonal bias $b_{m,\mathrm{seasonality}}$ is calculated by subtracting the $b_{m,\mathrm{annual}}$ from the difference of events grouped by month. Then, the root of the summed squares is built, therefore this bias has exclusively positive values:

$$b_{m,\mathrm{seasonality}} = \sqrt{\sum_{\mathrm{months}} \left( \frac{1}{N}\sum^{N} E_{m,\mathrm{monthly}} - \frac{1}{n}\sum^{n} E_{t,\mathrm{monthly}} - \frac{b_{m,\mathrm{annual}}}{12} \right)^2} \,. \tag{6}$$

– A bias for the inter-annual variability $b_{m,\mathrm{variability}}$ is calculated by subtracting the standard deviations of the yearly

events. Further, $b_{m,\mathrm{seasonality}}$ is subtracted from months with more foehn events and added to months with less foehn events compared to the training data. This information is obtained by the sign the subtraction yields before calculating the square in Eq. (6). The inter-annual variability therefore reads as

$$b_{m,\mathrm{variability}} = \mathrm{std}_{N}(E_{m,\mathrm{yearly}} \mp b_{m,\mathrm{seasonality}}) - \mathrm{std}_{n-1}(E_{t,\mathrm{yearly}}) \,. \tag{7}$$



Note that Bessel's correction of $n-1$ is used to account for the smaller set of years.

Then the biases of type $X$, are normalised with the mean absolute bias of all $M$ models, yielding the normalised bias

$$\tilde{b}_{m,X} = \frac{b_{m,X}}{\frac{1}{M}\sum_m |b_{m,X}|}, \tag{8}$$

for every model $m$ and type $X$. After that, the model weight $w_m$ is calculated by considering both individual model performance as well as independence from each other, as suggested by Knutti et al. (2017):

$$w_m = \frac{e^{-\frac{D_m^2}{\sigma_D^2}}}{1 + \sum_{j\neq i}^M e^{-\frac{S_{ij}^2}{\sigma_S^2}}}, \tag{9}$$

where $D_m = \sum_X |\tilde{b}_{m,X}|$ is describing the difference between model and observation, $S_{ij}$ is a measure for independence of individual models and the shape parameters $\sigma_S = \sigma_D = \frac{1}{M}\sum_m \left(\sum_X |\tilde{b}_{m,X}|\right)$, controlling the contribution of performance and independence, which is intended to be equal here (Sperna Weiland et al., 2021). By not weighting the sum resulting in $D_m$, it is ensured that each bias gets the same weight to the performance. $S_{ij}$ is calculated by computing the biases in Eq. (5) to (7) by replacing each instance of the training data $E_t$ with instances from a different model $E_j$, normalising the inter-model

differences according to Eq. (8) and summing over all biases, as for constructing $D_m$. Small values of $S_{ij}$ therefore indicate high dependency. The idea of this weighting method is that models agreeing poorly with observations (resulting in large $D_m$) and models largely duplicating existing models get smaller weights (Knutti et al., 2017). This weighting scheme is specifically suited for working with ensembles encompassing multiple initial conditions as it assigns smaller weights to members with identical initial conditions (Knutti et al., 2017) and ensembles, which were designed inter-dependent to each other, like the

CMIP5 ensemble (Brunner et al., 2019).

The final weights $\tilde{w}_m$ are normalised to 1 by dividing through the sum of weights

$$\tilde{w}_m = \frac{w_m}{\frac{1}{M}\sum_m w_m}. \tag{10}$$

The normalised weights $\tilde{w}_m$ are then used to derive the future trend, using the weighted mean and standard deviation calculated with the weighted variance for both spatial extents combined and widespread foehn only. To show the impact of

global warming more instructively, we use the global warming levels (GWL) 1.5°C, 2°C, 3°C and 4°C instead of displaying the trend with RCP scenarios (IPCC, 2022; James et al., 2017). This representation focuses on the impact of warming rather than the timing of when individual models reach those temperatures and allows models of different RCPs to contribute to one GWL. Which GWL corresponds to which period in the individual models is listed in appendix D.





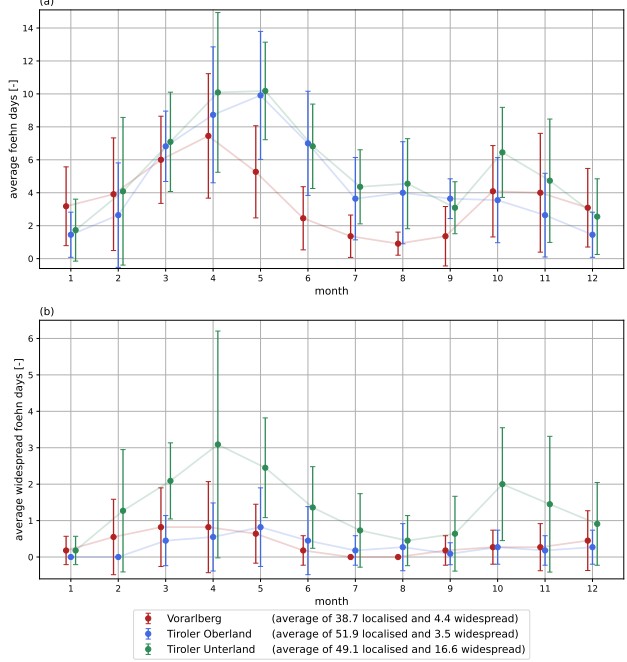

**Figure 3.** Seasonality in the training data, colour-coded by region. Displayed is the average number of foehn days per month for (a) all and (b) widespread events. The bars describe the standard deviation of the 11 years. The legend shows the average annual sum for every region.

## 3 Results

### 3.1 Training data

The application of OFC on selected station data yielded 11 years of daily training data with the seasonal behaviour visible in Fig. 3. It is also available publicly (Maier, 2024). The seasonality for all foehn events in panel a) shows similar relative seasonal behaviour of south foehn in the three regions, but different peak heights. This results in an average annual foehn occurrence of 11.8 % for Vorarlberg, 15.2 % for Tiroler Unterland and 17.9 % for Tiroler Oberland. The latter region shows the highest occurrence in widespread foehn events (4.5 %), as shown in panel b). While the results for the Tirol regions behave similarly for the first nine months, a second peak in foehn occurrence can be observed for Tiroler Oberland in panel a), which is lacking in Tiroler Unterland. This peak also coincides with a peak for widespread foehn in panel b), which is also lacking in the other two regions. The legend shows that while the number of localised foehn events is similar for the two Tirol regions, the number of widespread foehn events is higher by a factor of more than four for Tiroler Unterland compared to the other regions, as shown in panel b). The highest monthly standard deviation representing the inter-annual variability can be observed in April in Tiroler Unterland in both panels. Vorarlberg yields the smallest south foehn occurrence but more widespread events than Tiroler Oberland.





**Table 3.** Training scores for the six XGBoost models.

| Model | acc [%] [adpy] | TP [%] [adpy] | TN [%] [adpy] | ME [%] [adpy] | FA [%] [adpy] |
|---|---|---|---|---|---|
| Vorarlberg yes/no | 97.7 356.8 | 90.9 39.2 | 98.6 317.5 | 9.1 3.9 | 1.4 4.6 |
| Vorarlberg loc./wide. | 96.4 352.1 | 95.8 4.2 | 96.5 37.4 | 4.2 0.2 | 3.5 1.4 |
| Tiroler Oberland yes/no | 97.6 356.5 | 92.0 51.0 | 98.6 305.4 | 8.0 4.4 | 1.4 4.4 |
| Tiroler Oberland loc./wide. | 99.0 361.6 | 87.2 3.1 | 99.8 51.8 | 12.8 0.4 | 0.2 0.1 |
| Tiroler Unterland yes/no | 98.3 359.0 | 96.7 63.5 | 98.7 295.6 | 3.3 2.2 | 1.3 3.9 |
| Tiroler Unterland loc./wide. | 84.1 307.2 | 62.3 10.4 | 91.5 44.9 | 37.7 6.3 | 8.5 4.2 |

Loc. and wide. are short forms of localised and widespread. Adpy is an abbreviation for averave days
per year.

## 3.2 Training

The training scores of the six XGBoost models are visible in Table 3. All models achieve an accuracy of over 96 % with the
exception of the models that distinguishes between localised and widespread foehn in Tiroler Unterland, which achieves only
84.1 %. The balance between $ME$ and $FA$ is within 2 days per year for each model. When observing the most important
features in Table 4, the mean sea level pressure is identified as the most dominant feature by five out of six models. Only the
model distinguishing between localised and widespread foehn in Tiroler Oberland selects the wind direction in 500 hPa as
most dominant feature. The dominant features identified by all models, except for the one distinguishing between no foehn and
foehn in Tiroler Unterland, are located outside the corresponding region.

## 3.3 Weights

Figure 4 shows the normalised biases $\tilde{b}_{m,X}$ for (a) annual events, (b) seasonal accuracy and (c) inter-annual variability. Panel
(d) shows the summed absolute bias and (e) the resulting normalised weights $\tilde{w}_m$. For comparability, the same performance
analysis is applied on ERA5, which is visible in the first row of Fig. 4. The GCMs from the institutes ICHEC and MOHC show
a systematic negative bias in annual occurrence visible in Fig. 4a), independent of the selected RCM, whereas the models of



**Table 4.** Most important features of every XGBoost model. Features are given in abbreviations visible in Table 1 and are attributed to a location using the pressure level (plev), latitude (lat) and longitude (lon). Relative importance sums up to 1 for all features.

| Model | Feature [var, plev,lat,lon] | Relative Importance [-] |
|---|---|---|
| Vorarlberg yes/no | $\Delta p$, sfc , 47.00°, 12.75° | 0.24 |
| Vorarlberg loc./wide. | $\Delta p$, sfc , 47.00°, 13.75° | 0.10 |
| Tiroler Oberland yes/no | $\Delta p$, sfc, 47.00°, 13.75° | 0.11 |
| Tiroler Oberland loc./wide. | $wd$, 500 hPa, 48.00°, 11.75° | 0.22 |
| Tiroler Unterland yes/no | $\Delta p$, sfc, 47.00°, 11.7°5 | 0.06 |
| Tiroler Unterland loc./wide. | $\Delta p$, sfc, 47.25°, 14.00° | 0.08 |

Loc. and wide. are short forms of localised and widespread.

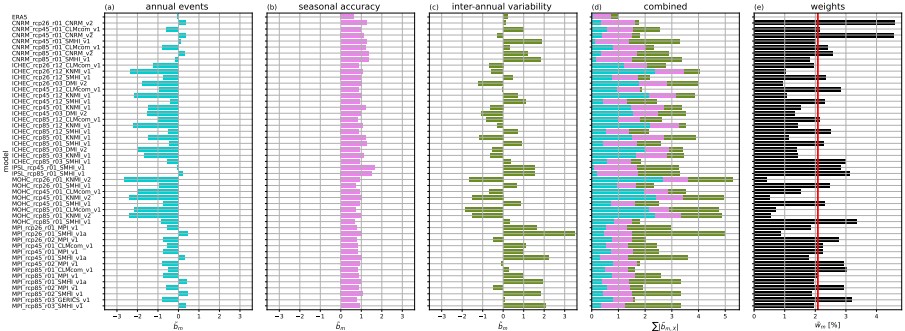

**Figure 4.** Bias and model weight composition. Displayed are the normalised biases $\tilde{b}_{m,X}$ of the historical period (1991-2020) compared to the training period (2011-2021) for every model for (a) the annual occurrence, (b) the seasonal accuracy and (c) the inter-annual variability of south foehn. Panel (d) displays the summed absolute bias. These biases result in the normalised weights in percent visible in panel (e). The averaged model weight of 2.08 % is displayed as red line. Model abbreviations are explained in appendix C.

the institutes CNRM, IPSL and MPI yield smaller biases with both signs. Figure 4b) shows that the EURO-CORDEX models yield similar biases for seasonal accuracy. The inter-annual variability in Fig. 4c) shows the largest spread of biases and a high inter-model variance. In the panels (a)-(d), a smaller bar represents smaller biases and therefore better performance. Figure 4e) shows the $\tilde{w}_m$ of the 48 models after using the weighting process suggested by Knutti et al. (2017), whereas larger bars 265 represent higher weightings. The weights range from 0.40 % to 4.59 % for the trend analysis in comparison to 2.08 % for every model without weighting (red line). A plot of the measure of independence $S_{ij}$ between individual EURO-CORDEX models can be found in appendix E.

Figure 5 shows the effect of the weighting on the seasonality in the historical period, summer across all regions in comparison to the training data and ERA5. It shows that the significant negative bias of annual occurrence is not eradicated by the weighting. 270 The inter-annual variability, represented by the standard deviation in the plot, is in comparable magnitude for ERA5 and the training data, whereas the EURO-CORDEX models, weighted or unweighted, provide standard deviations of approximately





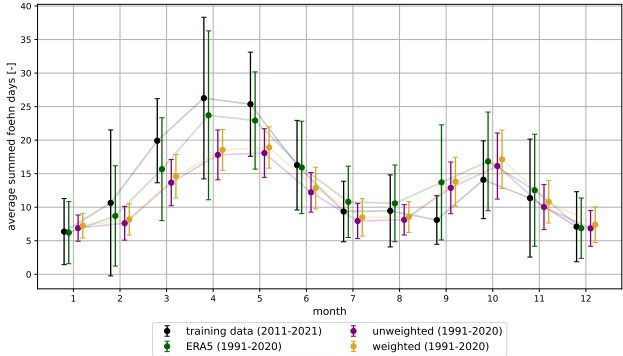

**Figure 5.** Summed seasonality of all regions in the historical period. Displayed is the average number of foehn days per month summed across all three regions for the weighted and unweighted EURO-CORDEX models in comparison to the training data and ERA5. The bars show the standard deviation of the 30 years (11 for the training data).

half the size. While the seasonality of the EURO-CORDEX models match the training data well as relative curve for most months, the models as well as ERA5 show more events in September and October than the training data.

### 3.4 Trend analysis

The weighted trend analysis in Fig. 6 shows the difference in foehn occurrence in the individual regions compared to the historical period, whereas the first column corresponds to all foehn events and the second only to widespread. Although the spread is large, a robust mean shift can be observed. A decrease in mean foehn occurrence for the two regions Tiroler Oberland in panel (c) and Tiroler Unterland in panel (d) to up to -5.6 %, depending on the GWL is visible. For example, this decrease corresponds to 62.2 foehn days in Tiroler Unterland at 4°C GWL in comparison to 65.7 in the historical period. Mean foehn
occurrence in Vorarlberg in panel (a) shows a decrease of around -2.5 % for all GWLs, except for the 4°C GWL, where occurrence achieves results similar to the historical period. The trend is therefore not as significant as for the Tirol areas. For widespread foehn occurrence, all three regions in panel (b,e,f) show an increase with GWL, with Vorarlberg yielding the most significant mean shift with up to 23.0 %. This would correspond to 5.4 days of widespread foehn on annual average compared to 4.4 in the historical period. A less significant trend is visible in the panels (e,f) for the Tirol regions.

For all regions, a seasonality shift in foehn occurrence can be observed in Fig. 7. Foehn probability increases with GWL in the spring months and even in February for 4°C GWL, while it decreases from July to September and also in October for 4°C GWL.

As suggested by Knutti et al. (2017), the same trend graphs using the unweighted ensemble can be found in appendix F. They generally agree with the displayed weighted trend of decreasing south foehn occurrence and increasing widespread events, with
some data points yielding slightly altered mean values or larger spread. Figure. 7 shows more agreement of monthly data points from the historical period and from the 1.5°C GWL compared to the unweighted counterpart.





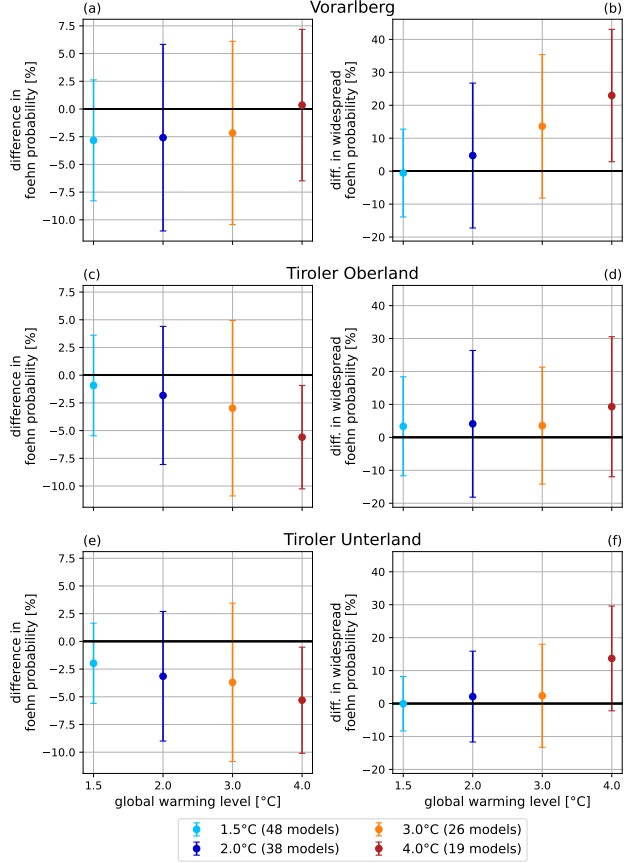

**Figure 6.** Foehn probability differences for all south foehn events (first column, panel (a,c,d)) and widespread events (second column, panel (b,e,f)) dependent on colour-coded GWL in °C compared to the historical period (1991-2020). The rows indicate different regions. The number of contributing models per GWL is visible in brackets in the legend.

## 4 Conclusions

In this study, CMIP5 EURO-CORDEX models are tested in their ability to produce south foehn over the eastern Alps. This was done by applying the state-of-the-art machine learning technique XGBoost for three regions over western Austria, namely

Vorarlberg, Tiroler Unterland and Tiroler Oberland. First, 11 years of daily training data were generated by performing OFC on semi-automated weather station data. The resulting south foehn occurrence of 17.9 % for Tiroler Unterland is in compareable range to results of other studies in this geographical area, ranging from 15.7 to 19.7 % for a single station in the Wipp Valley depending on the detection method used (Plavcan et al., 2014). The results show differences in occurrence frequency in Vorarlberg and the Tirol areas. We therefore conclude that Zängl et al. (2004) were correct in attributing different flow characteristics

for the Rhine valley in Vorarlberg and the Wipp Valley for Tiroler Unterland and that separating the regions Vorarlberg and Tirol is justified. Furthermore, the more than four times higher probability of widespread events in Tiroler Unterland compared



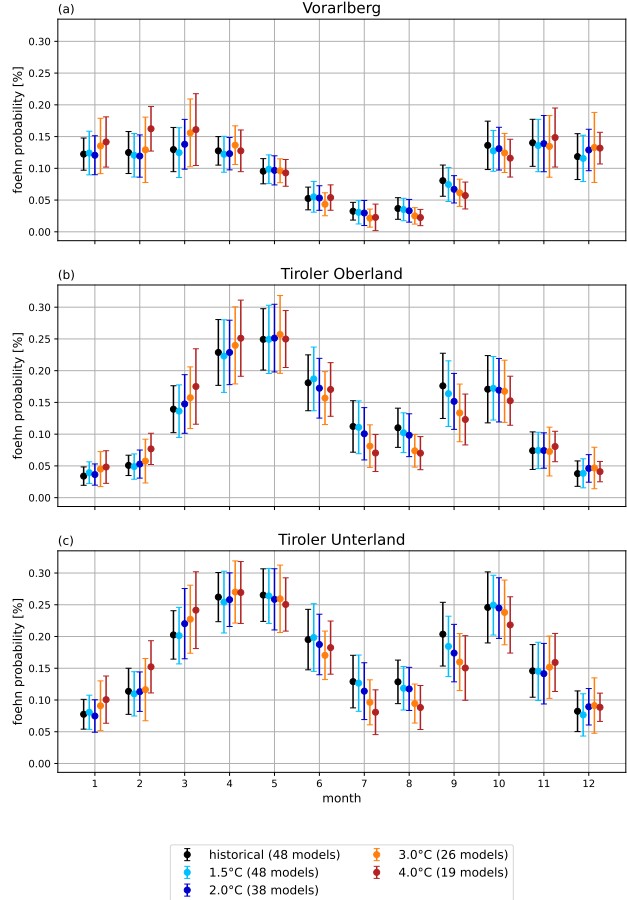

**Figure 7.** Seasonality trend for the probability of south foehn in % for the regions (a) Vorarlberg, (b) Tiroler Oberland and (c) Tiroler Unterland, dependent on GWL in °C. The number of contributing models per GWL is visible in brackets in the legend.

to Tiroler Oberland manifests the special role of the Wipp Valley in providing ideal conditions for foehn. Separation of Tiroler Oberland and Tiroler Unterland is therefore also justified.

Together with meteorological data from ERA5, one XGBoost model per region and spatial extent (localised and widespread) was trained. Although fewer meteorological input parameters compared to other studies were used (Mony et al., 2021), satisfying training accuracies of above 96 % for five out of six models were achieved by performing an intensive hyperparameter grid search. We found that the limited selection of available variables in EURO-CORDEX models does not hinder a machine learning approach in reaching similar accuracies of around 97.5 % observed in other studies (Mony et al., 2021). Only the model for distinguishing between localised and widespread foehn in Tiroler Unterland yielded an accuracy of 84.1 %. We attribute

the lower accuracy to the special role of the Wipp Valley located in the region. Due to its ideal shape and perpendicular orientation to the Eastern Alps' main crest, the valley enables foehn to descent in the valleys more likely, making the distinguishing



between localised and widespread events less recognisable in synoptic patterns, resulting in more missed events. The balance between missed events and false alarms is within two days for every model, indicating, that the models succeeded to train on the imbalanced data set to a satisfying degree. In five out of six cases, mean sea level pressure gradients are identified as the

most important features, attributing it as the most important driver for foehn in accordance to Richner and Hächler (2013). The most important features are not located in the corresponding regions in five out of six cases. This may indicate that the input variable selection still could be reduced due to the high correlation of adjacent grid cell values.

The trained XGBoost models were then applied to the OEKS-15 selection of CMIP5 EURO-CORDEX climate projections. By analysing the annual occurrence, seasonal accuracy and inter-annual variability of the historical period (1991-2020) to the

training data (2011-2021), the performance of individual models compared to each other and to ERA5 was analysed. No model performed better than the reanalysis ERA5, as expected. Although ERA5 itself struggles to accurately capture the seasonality, most EURO-CORDEX achieve comparable seasonal accuracy. The annual bias is heavily influenced by the driving GCM and shows a systematic negative bias for the GCMs ICHEC-EC-EARTH and MOHC-HadGEM2-ES. Their poor performance is in contrast to other studies, where ICHEC-EC-EARTH performed best for extratropical cyclones in the cool season over

the western Atlantic (Colle et al., 2013) and for hydrological impacts in Europe (Sperna Weiland et al., 2021) and MOHC-HadGEM2-ES ranked in the middle tier in both studies. Both also perform well in producing European extratropical cyclones (Zappa et al., 2013). However, they also show significant cold biases in winter and summer in the (greater) Alpine region and most EURO-CORDEX realisations of these two GCMs suggest a wet bias and a negative mean sea level pressure bias (Vautard et al., 2021; Coppola et al., 2021). While this is a first hint in what could hinder these models in reproducing annual foehn

occurrence, we recommend further investigation of the dynamics of the EURO-CORDEX ensemble over the alpine region.

The performance analysis and weighting process for the trend analysis show that EURO-CORDEX RCMs significantly differ in being able to produce south foehn over western Austria, as indicated by the bias analysis. The weighting process cannot eliminate the overall negative annual bias in foehn occurrence or increase the inter-annual spread of events compared to the training data. It succeeds in reducing the spread and predicting a more robust climate change signal in the future foehn

trend analysis compared to the unweighted analysis, which was the desired outcome. As CMIP5 models were not designed to perform independently from each other as developers might have shared ideas, code and used identical components or models branched from an identical predecessor, we see the weighting process as justified as model independence was also included in the weighting process (Knutti et al., 2013; Sanderson et al., 2015; Lorenz et al., 2018; Abramowitz et al., 2019).

The weighted trend analysis revealed a slight, yet robust trend of decreasing foehn occurrence in Tiroler Oberland and

Tiroler Unterland depending on the GWL. Vorarlberg does not show a trend, but rather a stable value of smaller frequency, except for the 4°C GWL, where it shows compareable values to the historical period. A clear positive trend with rising GWL when observing widespread foehn is visible for all regions, especially for Vorarlberg. For all regions, we observed a seasonality shift in foehn occurrence towards the earlier spring months, with a decrease in summer and also in October at the higher GWLs. This confirms other predictions for south foehn seasonality in Altdorf, Switzerland (Mony et al., 2021).

The limiting factors of the study are discussed here. First, the different spatial resolutions of ERA5 and EURO-CORDEX yielded that the finer EURO-CORDEX models had to be regridded onto the ERA5 grid as training was done on the grid of



the reanalysis, resulting in a loss of spatial variability. Second, the training data were created by using hard thresholds for the definition of foehn, which originate from our current knowledge and understanding of foehn formation. Changing those thresholds would lead to different training data. Third, the machine learning techniques are trained to detect the synoptic patterns leading to foehn. If global warming would result in foehn-producing synoptic patterns of different strength and range, the approach would not be able to capture it. Finally, the selection of a performance metric, the weighting process and its shape parameters introduce subjectivity to the trend analysis of climate projections, which could be further mitigated by cross validation or a perfect model setup (Knutti et al., 2017).

The training process using ERA5 and the approach of weighting climate projections by the mentioned performance parameters could be applied to different foehn-prone regions. We also recommend repeating the study once regional models forced with CMIP6 GCMs become publicly available, given the anticipated improved sensitivity of the new model generation (Palmer et al., 2021). Furthermore, the study raises the question, what the physical drivers behind the altered foehn occurrence and seasonality are and how synoptic patterns are modified. It also raises the question of which processes the poorly performing CMIP5 projections are not able to capture adequately. An approach to address both questions would involve classifying weather conditions that lead to foehn and tracking their changes in the wake of climate change. The identification of the physical driver is beyond the scope of this study and is planned to be conducted in future research.

*Code and data availability.* The weather station data is available in the GeoSphere Austria data hub (URL: https://data.hub.geosphere.at/dataset/synop-v1-1h). ERA5 is available in the COPERNICUS climate data store (URL: https://cds.climate.copernicus.eu/cdsapp#!/dataset/reanalysis-era5-complete?tab=overview). The EURO-CORDEX models are available over the ESGF infrastructure (e.g. https://esgf-data.dkrz.de/search/cmip5-dkrz/). The foehn training data created in this study is publicly available (DOI: https://doi.org/10.5281/zenodo.10478610). Code for the analysis and visualisation is written in Python and can be obtained from the authors on request.

*Author contributions.* All authors contributed to conceptualisation and methodology; PM created the software and carried out the data analysis and visualisation; PM is responsible for data curation; PM, FL and HF validated the results; PM wrote the manuscript draft; FL, TK and HF reviewed and edited the manuscript; HF supervised the project.

*Competing interests.* The authors declare that they have no conflict of interest.

*Acknowledgements.* The authors want to thank Mathias Rotach, Michael Sprenger and Lukas Jansing for fruitful discussions. We also want to thank the Austrian Federal Ministry of Agriculture, Regions and Tourism, the Austrian federal states Tirol and Vorarlberg and the European



Agricultural Fund for Rural Development for funding the project, in which this study originated. We also want to aknowledge, that we used the AI tools ChatGPT and Phind for proofreading the final manuscript and gathering literature references.





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



## Appendix A: Station locations

**Table A1.** Selected station observations from GeoSphere Austria (2020).

| Region | Name | Crest or valley | Latitude [°] | Longitude [°] | Elevation [m] |
|---|---|---|---|---|---|
| Vorarlberg | Valluga[†] | Crest | 47.157500 | 10.212778 | 2805 |
| | Bludenz | Valley | 47.147778 | 9.829444 | 565 |
| | Brand | Valley | 47.157500 | 10.212778 | 1029 |
| | Dornbirn | Valley | 47.432500 | 9.725556 | 407 |
| | Feldkirch | Valley | 47.271111 | 9.609722 | 439 |
| | Gaschurn | Valley | 46.985833 | 10.028611 | 976 |
| | Schoppernau | Valley | 47.311389 | 10.017778 | 839 |
| Tiroler Oberland | Patscherkofel | Crest | 47.208889 | 11.462222 | 2251 |
| | Galtuer | Valley | 46.968056 | 10.185556 | 1587 |
| | Imst | Valley | 47.236944 | 10.742222 | 773 |
| | Landeck | Valley | 47.133333 | 10.566667 | 796 |
| | Reutte | Valley | 47.494444 | 10.715278 | 842 |
| | St. Anton/Arlberg | Valley | 47.131389 | 10.266667 | 1304 |
| | St. Leonhard/Pitztal | Valley | 47.027222 | 10.865556 | 1454 |
| | Umhausen | Valley | 47.139167 | 10.928889 | 1035 |
| Tiroler Unterland | Patscherkofel | Crest | 47.208889 | 11.462222 | 2251 |
| | Achenkirch | Valley | 47.532222 | 11.705278 | 905 |
| | Alpbach | Valley | 47.396389 | 11.940556 | 929 |
| | Brenner | Valley | 47.007222 | 11.510833 | 1412 |
| | Innsbruck University | Valley | 47.260000 | 11.384167 | 578 |
| | Jenbach | Valley | 47.388889 | 11.758056 | 530 |
| | Kufstein | Valley | 47.575278 | 12.162778 | 491 |
| | Mayrhofen | Valley | 47.162500 | 11.851389 | 640 |
| | Steinach | Valley | 47.098333 | 11.466111 | 1036 |

[†] Valluga pressure values are calculated using the pressure values from Galzig and the barometric height formula with a linear temperature gradient. Galzig was therefore removed as valley station.



## Appendix B: XGBoost training hyperparameters

**Table B1.** XGBoost hyperparameter selection for the best accuracy per XGBoost model.

| Model | Evaluation metric | Gamma | Learning rate | Maximum depth | Minimum child weight | Objective | Regression lambda |
|---|---|---|---|---|---|---|---|
| Vorarlberg yes/no | error | 1.5 | 0.3 | 9 | 10 | binary:logitraw | 2.5 |
| Vorarlberg loc./wide. | logloss | 1 | 0.8 | 7 | 1 | binary:logitraw | 2.5 |
| Tiroler Oberland yes/no | error | 1.5 | 0.3 | 9 | 10 | binary:logitraw | 2.5 |
| Tiroler Oberland loc./wide. | logloss | 1 | 0.8 | 7 | 1 | binary:logitraw | 2.5 |
| Tiroler Unterland yes/no | error | 1 | 0.3 | 9 | 5 | binary:logitraw | 1.5 |
| Tiroler Unterland loc./wide. | error | 1 | 0.5 | 5 | 10 | binary:logitraw | 2.5 |

Loc. and wide. are short forms of localised and widespread.



## Appendix C: EURO-CORDEX model abbreviations

The name of a EURO-CORDEX model is specified by

– GCM - originating institute and name,

– RCP scenario,

– Ensemble run,

– RCM - originating institute and name and

– downscaling version,

usually separated by underscores. We decided to shorten the names by using the originating institute names for the GCM
and RCM and by limiting the ensemble run to the first digit(s). A shorted name is therefore comprised as GCM-institute_RCP-scenario_ensemble-run_RCM-institute_downscaling-version. The following abbreviations are used:

**Table C1.** Model Abbreviations.

|  | full | abbreviated |
|---|---|---|
| GCM | CNRM-CERFACS-CNRM-CM5 | CNRM |
|  | ICHEC-EC-EARTH | ICHEC |
|  | IPSL-IPSL-CM5A-MR | IPSL |
|  | MOHC-HadGEM2-ES | MOHC |
|  | MPI-M-MPI-ESM-LR | MPI |
| Ensemble runs | r1i1p1 | r01 |
|  | r2i1p1 | r02 |
|  | r3i1p1 | r03 |
|  | r12i1p1 | r12 |
| RCM | CLMcom-CCLM4-8-17 | CLMcom |
|  | CNRM-ALADIN63 | CNRM |
|  | DMI-HIRHAM5 | DMI |
|  | GERICS-REMO2015 | GERICS |
|  | KNMI-RACMO22E | KNMI |
|  | MPI-CSC-REMO2009 | MPI |
|  | SMHI-RCA4 | SMHI |



## Appendix D: GWL periods

**Table D1.** Periods, in which an individual model is reaching certain GWLs. The periods are independent of the chosen RCM and downscaling version.

| GCM_RCP_Ensemble-run | GWL [°C] | | | |
| | 1.5 | 2.0 | 3.0 | 4.0 |
| --- | --- | --- | --- | --- |
| CNRM-CERFACS-CNRM-CM5_rcp26_r1i1p1 | 2034-2053 | | | |
| ICHEC-EC-EARTH_rcp26_r3i1p1 | 2014-2033 | | | |
| ICHEC-EC-EARTH_rcp26_r12i1p1 | 2014-2033 | | | |
| MOHC-HadGEM2-ES_rcp26_r1i1p1 | 2014-2033 | | | |
| MPI-M-MPI-ESM-LR_rcp26_r1i1p1 | 2013-2032 | | | |
| MPI-M-MPI-ESM-LR_rcp26_r2i1p1 | 2007-2026 | | | |
| CNRM-CERFACS-CNRM-CM5_rcp45_r1i1p1 | 2028-2047 | 2049-2068 | | |
| ICHEC-EC-EARTH_rcp45_r1i1p1 | 2013-2032 | 2035-2054 | | |
| ICHEC-EC-EARTH_rcp45_r3i1p1 | 2013-2032 | 2035-2054 | | |
| ICHEC-EC-EARTH_rcp45_r12i1p1 | 2013-2032 | 2035-2054 | | |
| IPSL-IPSL-CM5A-MR_rcp45_r1i1p1 | 2007-2026 | 2024-2043 | 2068-2087 | |
| MOHC-HadGEM2-ES_rcp45_r1i1p1 | 2019-2038 | 2034-2053 | 2069-2088 | |
| MPI-M-MPI-ESM-LR_rcp45_r1i1p1 | 2013-2032 | 2035-2054 | | |
| MPI-M-MPI-ESM-LR_rcp45_r2i1p1 | 2010-2029 | 2032-2051 | | |
| CNRM-CERFACS-CNRM-CM5_rcp85_r1i1p1 | 2021-2040 | 2036-2055 | 2058-2077 | 2078-2097 |
| ICHEC-EC-EARTH_rcp85_r1i1p1 | 2010-2029 | 2026-2045 | 2052-2071 | 2073-2092 |
| ICHEC-EC-EARTH_rcp85_r3i1p1 | 2010-2029 | 2026-2045 | 2052-2071 | 2073-2092 |
| ICHEC-EC-EARTH_rcp85_r12i1p1 | 2009-2028 | 2025-2044 | 2051-2070 | |
| IPSL-IPSL-CM5A-MR_rcp85_r1i1p1 | 2006-2025 | 2021-2040 | 2041-2060 | 2057-2076 |
| MOHC-HadGEM2-ES_rcp85_r1i1p1 | 2014-2033 | 2026-2045 | 2045-2064 | 2062-2081 |
| MPI-M-MPI-ESM-LR_rcp85_r1i1p1 | 2008-2027 | 2028-2047 | 2052-2071 | 2072-2091 |
| MPI-M-MPI-ESM-LR_rcp85_r2i1p1 | 2007-2026 | 2023-2042 | 2050-2069 | 2071-2090 |
| MPI-M-MPI-ESM-LR_rcp85_r3i1p1 | 2011-2030 | 2026-2045 | 2050-2069 | 2071-2090 |




## Appendix E: EURO-CORDEX model independence

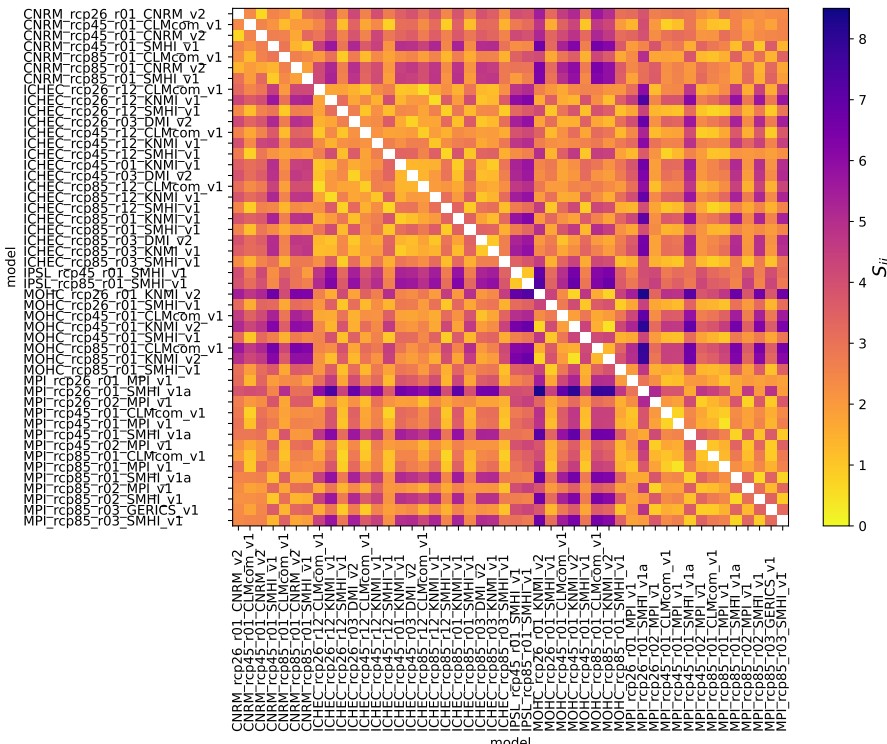

**Figure E1.** Normalised EURO-CORDEX model independence measure $S_{ij}$. The axes indicate different EURO-CORDEX models, the colourbar indicates how dependent individual models are on each other, with higher values suggesting lower dependency as suggested by Eq. (9). The plot is mirrored along the diagonal, which is displaying $S_{i,j=i} = 0$.



## Appendix F: Unweighted trends

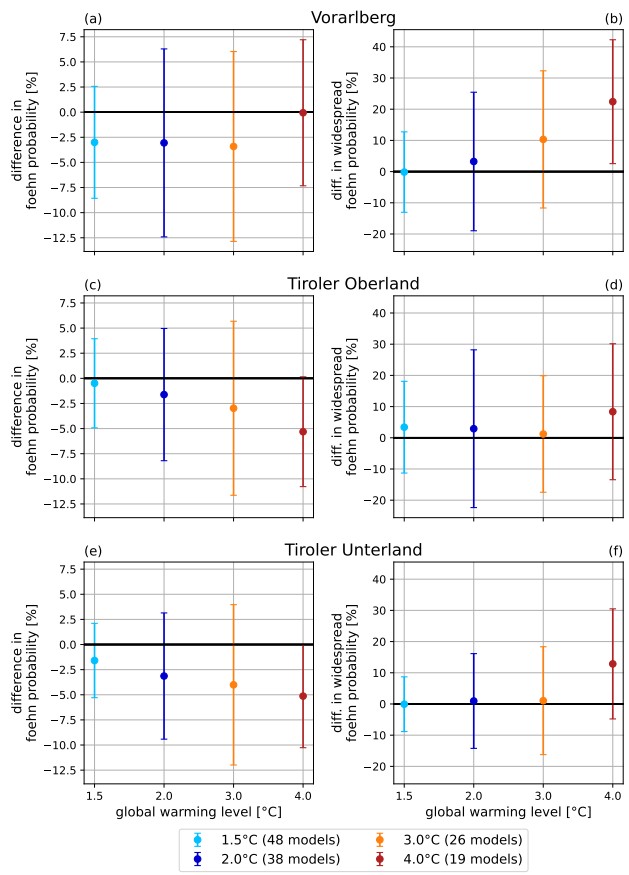

**Figure F1.** Same as Fig. 6, but with the unweighted ensemble.





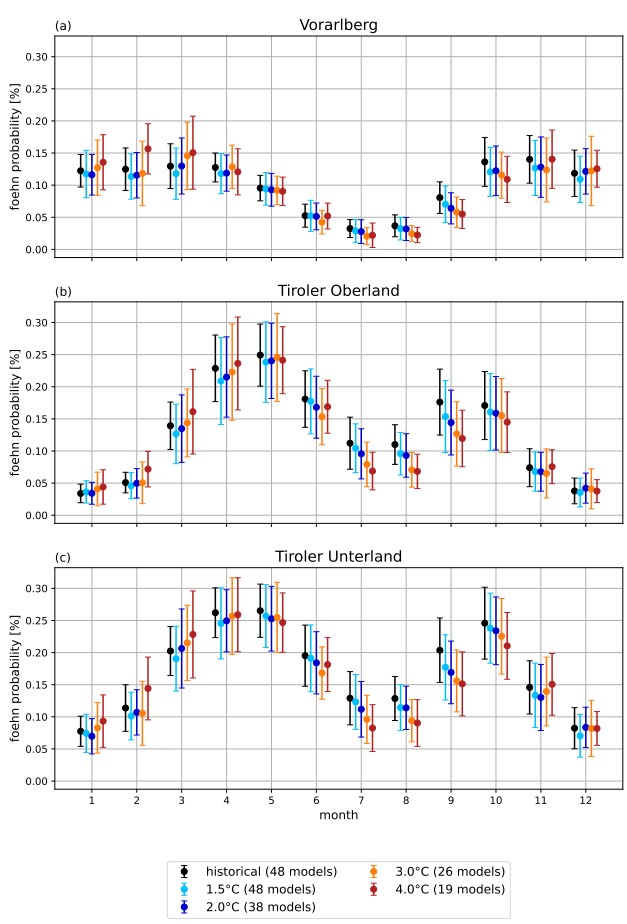

**Figure F2.** Same as Fig. 7, but with the unweighted ensemble.