# Peer review of "Analysing CMIP5 EURO-CORDEX models in their ability to produce south foehn and the resulting climate change impact on frequency and spatial extent over western Austria"

_EGUsphere, 2024_

## Referee Comment (RC1)

**Review for**

**"Analysing CMIP5 EURO-CORDEX models in their ability to produce south foehn and the resulting climate change impact on frequency and spatial extent over western Austria"**

**Synthesis:**

The present study addresses a relevant topic, namely potential changes in the occurrence of foehn events under a changing climate, thereby focusing on western Austria in the Alps. Following a methodology similar to that of a previous study, the authors employ six XGBoost models to indirectly address this question. These models are trained using ERA5 variables in order to predict the occurrence of both localized and widespread foehn events across three subregions, with station-based foehn detection data providing the training labels. Subsequently, the machine-learning models are applied to CMIP5-CORDEX climate simulations for present-day and future climate scenarios. The authors find a shift in the seasonality of foehn events under future climate, with foehn becoming more frequent in spring and less frequent in autumn.

The manuscript tackles a relevant research gap and chooses a suitable methodological approach. The study would therefore be a valuable addition to the existing foehn literature, particularly given the limited understanding of how the frequency of foehn events may be influenced by global warming. However, there is a need for a more detailed explanation of the training and validation process of the XGBoost models, as highlighted in the first major comment. Additionally, there is room for refinement in both the structure and content of the manuscript: currently, the methods section is lengthy in contrast to the relatively concise and descriptive results section. Expanding the discussion of the results, as suggested in the second major comment, would significantly enhance the manuscript. Furthermore, there are many minor comments to be addressed that would further improve readability, consistency, and the overall quality of the manuscript. For these reasons, I recommend that the authors address these concerns if the manuscript is to be considered for publication in WCD.

**Major comments:**

1. **Approach to training and validation:** According to the reviewers' understanding, the authors opted not to partition the ERA5 and the OFC data into distinct training and test sets. Typically, machine learning models are trained on a subset of the data, while another subset remains unseen for the model and is reserved for testing and performance evaluation (see, e.g., Mony et al., 2021). Consequently, comparing performance metrics with other studies might be somewhat misleading, as the evaluation relies on the time period used for model training. I understand that the authors conduct an indirect performance evaluation by comparing the seasonal foehn frequency derived from OFC data in the time period of 2011-2021 to that of ERA5 and EURO-CORDEX data derived from the period 1991-2020. Nevertheless, I am curious how the authors

intend to demonstrate that the current approach does not lead to overfitting and consequently render the results less generalizable, particularly considering the differences we see between OFC and ERA5, but especially between OFC and EURO-CORDEX models in Figure 5. The temporal mismatch of the time period where the OFC data is available (2011-2021) compared to the historical time period used for indirect validation (1991-2020) makes it even more challenging to judge whether the XGBoost models have been successfully trained and applied to the present-day EURO-CORDEX simulations. Please comment on these aspects and discuss them in the manuscript.

2. The authors have made a clear distinction between the descriptive results and their interpretation, which is a valid option. However, I believe that **a more extensive discussion of the results is necessary, extending beyond what is currently presented in the conclusions.** Whether this extended discussion is incorporated at the end of each individual results section, integrated within more comprehensive conclusions, or presented as a distinct discussion section is up to the authors. However, in my opinion, such an addition would create a more equal balance between the actual results and discussion within the paper, especially compared to the current methods section. I give several suggestions for additional interpretation and discussion of the results below (further suggestions are provided in the minor comments as well):

    a) ***Interpret and discuss the seasonal foehn occurrence in the three regions for the present-day period according to OFC***: Figure 3 presents intriguing results and could be discussed in more detail. Is this the first "climatology" of foehn occurrence in Vorarlberg and Tiroler Oberland? The authors could highlight this more explicitly if no such prior publication exists. Additionally, several open questions arise when looking at the differences in the seasonal cycle of foehn between the three regions: Why does Vorarlberg feature substantially less foehn days during most months compared to the Tirol regions? Why do widespread events appear to be more prevalent in Tiroler Unterland compared to the other regions? Considering the size of the study area, one would anticipate that foehn events occur under similar synoptic weather conditions. The authors could validate this assumption by computing ERA5 composites for the mean synoptic situation during foehn days in 2011-2021 across the three regions.

    Local effects, rather than synoptic differences, are probably responsible for the above-mentioned differences between the three regions. The authors mention the north-south valley axis of the Wipp Valley as a factor favoring foehn breakthrough (L. 38 and L. 302). Are none of the other stations located in a valley that is also north-south oriented? It is worth noting that several other factors influence the foehn climatology at specific stations. Foehn is more likely to occur at stations that are closer to the main Alpine crest, both horizontally and vertically (e.g., Gutermann et al., 2012). Additionally, a deep incision in the crest, such as the Brenner gap, favors foehn penetration, especially during shallow foehn episodes that are primarily driven by the overflow of colder air from the southern to the northern side (e.g., Jansing et al., 2022). Another crucial factor to consider is a station's tendency to cold-air pooling. During the colder months, the foehn flow at lowerelevation stations in valleys often fails to reach the ground due to the stable valley air mass hindering its breakthrough (e.g., Drobinski et al., 2007; Haid et al., 2022). How strong is this effect at the different stations across the three regions? I am particularly surprised by the higher occurrence of widespread events in the Tiroler Unterland compared to other regions, as I had previously assumed that stations in the Inn Valley (such as Innsbruck, Jenbach, and Kufstein) are prone to cold-air pooling, especially in autumn and winter. Moreover, it is worth noting that local cold-air pools can pose challenges for the accurate prediction of foehn occurrence by the XGBoost models, as foehn-prone conditions might not lead to foehn breakthrough under such circumstances. I encourage a discussion of these factors in relation to the seasonal foehn occurrence diagnosed from OFC data, as well as their implications for the XGBoost predictions.

b) ***Provide a more comprehensive description and interpretation of the training results***: The authors might consider incorporating metrics such as the correct alarm ratio and probability of detection (as suggested in minor comment 16), as the accuracy metric may present an overly optimistic view on the model performance given the imbalanced dataset. This would enable a more precise evaluation of the performance discrepancies among the XGBoost models. For instance, the correct alarm ratio for Vorarlberg loc. / wide., as well as the probability of detection and the correct alarm ratio for Tiroler Unterland loc. / wide. seem to be lower compared to other metrics.

Moreover, the authors could consider showing the importance of additional features, or at least indicate whether they hold nearly equal importance or are considerably less significant compared to the most important feature (Table 4). Additionally, Figure 5 could be expanded to multiple panels to illustrate the results for all three regions and both categories (localized/widespread). This represents one of the main results of the study. Presently, with the existing Figure 5, readers are unable to discern how ERA5, the weighted, and the unweighted EURO-CORDEX models capture the seasonality across different regions and categories. The manuscript would benefit significantly from including these results and discussing them within the text.

c) ***Revisit the description and interpretation of the XGBoost models applied to ERA5 and the EURO-CORDEX models for the historical period***: Unfortunately, there is a temporal discrepancy between the time periods depicted for the training data (2011-2021) based on OFC and the predictions from the XGBoost models (1991-2020). How does the ERA5 seasonality look like when only considering the training period from 2011-2021? The same question is also valid for the XGBoost predictions using EURO-CORDEX data. This comment is also related to comment a) further above.

**Minor comments:**

1. The title ("… produce south foehn …") is somewhat misleading in the sense that the EURO-CORDEX models do not actually simulate foehn flows that resemble actual foehn flow

characteristics, since their resolution, as the authors correctly state in their manuscript, is too coarse to resolve individual foehn valleys. These models only represent the synoptic (and mesoscale) conditions that are typically associated with the occurrence of foehn flows in the Alps. Consider rephrasing the title of the manuscript.

2. L. 9: Similar to the first minor comment, the EURO-CORDEX models do not "produce south foehn" in the sense that these flows are resolved in the models. Instead, foehn is diagnosed using the atmospheric fields that represent the typical synoptic to mesoscale weather situations during foehn and non-foehn conditions.

3. L. 36ff: Please consider expanding this sentence. It is, in my opinion, oversimplifying the foehn research in the Alpine region, which is not limited to two field campaigns. If you are referring to the MAP field campaign, it would be more accurate to cite Mayr et al. (2007) and Drobinksi et al. (2007) to adequately represent the MAP findings concerning foehn flows. Furthermore, there has been a recent, more local-scale campaign focusing on foehn-cold-air-pool interactions (Haid et al., 2020, 2022.). Moreover, there exists a range of recent publications that focus on the representation of Alpine foehn flows in mesoscale to large-eddy simulations, aiming to understand the physical processes governing foehn flows and to improve the quality of foehn simulations (e.g., Seibert et al., 2000; Würsch and Sprenger, 2015; Miltenberger et al., 2016; Umek et al., 2021; Jansing and Sprenger, 2022; Jansing et al., 2022; Saigger and Gohm, 2022; Umek et al., 2022; Jansing et al., 2024; Tian et al., 2024). Consider incorporating this important aspect of foehn research in an additional statement and referring to selected publications.

4. L. 40: The descent of foehn in the framework of hydraulic theory does not occur due to the flow transition into critical or supercritical flow, but due to cross-Alpine density differences (buoyancy-driven, see, e.g., Mayr et al., 2007).

5. L. 41ff: Forecasting foehn using statistical methods extends further back in time. Widmer (1966) and Courvoisier and Gutermann (1971) already developed an index to predict foehn at Altdorf, Switzerland. Moreover, Dürr (2008) also developed an objective foehn classification method using station data.

6. L. 46: This is actually not true, there are by now regional climate projections with a spatial resolution down to 2.2 km (e.g., Ban et al., 2021). In these models, one could actually at least partially resolve the foehn flows and thus explicitly investigate foehn flows under future climate.

7. L. 51: Given the similarity in approach, it would be beneficial to summarize the main findings of Mony et al. (2021) in one or two sentences. This would sharpen the research addressed in the study, specifically the investigation of future foehn occurrence over Austria, a topic that has not been explored previously. For instance, it has remained uncertain whether the conclusions drawn by Mony et al. (2021) could be replicated with other climate models and applied to foehn regions in the Eastern Alps.

8. L. 59-73: A substantial paragraph of the introduction is devoted to a discussion of the advantages and disadvantages of ensemble weighting. I wonder if at least part of this discussion would be better placed in the methods section. The current structure of this paragraph overemphasizes this particular method and focuses too little on the actual aims of the study. Therefore, the author may consider restructuring this part of the introduction to better align with the overall focus of the research.

9. Fig. 2: Nice flowchart! Why not drawing an arrow from "11 years of daily training data" to "Training of two XGBoost models per region" as well?

10. Table 1: How is the model main ridge defined? Please specify this.
11. Table 1 vs. L. 97-99: Table 1 is not fully consistent with the text; for instance, the cross-Alpine potential temperature gradient is not mentioned in the text. Additionally, could the authors provide further insight into their rationale behind selecting the cross-Alpine pressure gradient within ± 1° latitude relative to the model's main ridge as a feature (only 37 grid points), compared to the inclusion of every grid point for the geopotential difference on 500 hPa (925 grid points)? The same question applies to the cross-Alpine potential temperature gradient. Moreover, the authors might consider elaborating on why they opted for the relative humidity difference with respect to the monthly mean as an additional feature, especially since it is the only variable where an anomaly is considered. It would be beneficial for readers to gain a deeper understanding of the motivation behind the selection of the feature matrix input.
12. L. 100-101: This is actually not the study domain (not the red box in Fig. 1a), but the domain considered to extract features from ERA5.
13. L. 110: Can the authors specify how many models are included in this OEKS15 selection?
14. L. 117ff: In my opinion, this paragraph could be shifted to the beginning to the next paragraph, as it is directly related to the OFC used to generate the training labels. If done so, I would also suggest to rename section 2.1, e.g., to "2.1 Reanalysis data and climate projections".
15. L. 141-142: Judging from Fig. 4b, Vorarlberg does not feature a less dense network of stations, but simply is the smaller region.
16. L. 179-182: As the authors correctly point out, the accuracy (acc) is not a suitable metric for a highly unbalanced dataset like the one presented here. They could therefore consider to also evaluate the correct alarm ratio (TP / (TP + FA)) and the probability of detection (TP / (TP + ME)), which would give the reader a more comprehensive picture of the model performance on the training dataset.
17. L. 199ff / Eqs. 5-7: As the authors compare two different time periods (1991-2020 vs. 2011-2021), this correction is only valid if the annual frequency, the seasonality and the inter-annual variability in the period 1991-2010 do not differ too much from the period 2011-2021. Unfortunately, this cannot be tested with the training data used in this study. It should be noted that years with an unusually high or low foehn frequency often co-occur over several years (e.g., Richner et al., 2014). Did the authors try a correction where they calculated the biases using only the period 2011-2020 (i.e., N = 10 instead of N = 30)?
18. L. 205ff: Why is the seasonality bias ($b_{m,seasonality}$) either added or subtracted from the monthly foehn events before computing the standard deviation?
19. Section 2.4: This section is rather lengthy, describing the methodological procedure of weighting of the different simulations. It therefore appears imbalanced, especially compared to the relatively short results section. The authors might consider shifting part of this explanation into an appendix (e.g., appendix D).
20. L. 235, L. 248, L. 256, L. 274: The section titles reflect the methodology used rather than the results they present. Why not rename them, for example, to "3.1 Present-day foehn climatology based on station data and OFC", "3.2 Model performance for present-day climate" (merge 3.2 and 3.3 as 3.2 is very short anyway), "3.3 Foehn occurrence under future climate"? These are just suggestions, but consider renaming the sections to make them more intuitive.
21. Figure 3: The caption explains that the uncertainty bars show the standard deviation over the 11 years. In my opinion, it would be more intuitive to adjust the ranges to the minimum and

maximum, or to the 10$^{th}$ and 90$^{th}$ percentiles. The current visualization is misleading as there are certainly not less than 0 foehn days in a given month.

22. Figure 3: I suggest to show the average localized foehn days in a third panel and the temporal evolution (i.e., the inter-annual variability) in a fourth panel. This would allow the reader to get a more complete picture of the contrasting behavior of the three regions (e.g., Tiroler Unterland has less localized but more widespread events than Tiroler Oberland – really surprising to me!)

23. L. 239-242: The text is not consistent with the figure. The figure shows that Tiroler Unterland has the highest overall annual foehn frequency and a secondary maximum in autumn due to widespread events. In the text, however, the authors claim that the Tiroler Oberland has the highest annual foehn frequency (17.9%) and the highest number of widespread events. Which is correct?

24. L. 281 and L. 284: Please be careful when talking about "significant trends", especially in the context of statistical trend analysis. Did the authors test the trends for significance using a statistical method?

25. Figure 7: Given the different trends for localized and widespread events in Figure 6, it would be interesting to see Figure 7 separately for localized and widespread events. If the authors feel that this is not appropriate in the main text, they could include it in the appendix (or a supplement).

26. L. 298-303: Here the authors could discuss the other possible explanations for the regional differences (see major comment 2a).

27. L. 307-308: Consider comparing the correct alarm ratio and the probability of detection with Mony et al. (2021). Note that the comparison is somewhat misleading, as the authors did not apply the XGBoost models to a test dataset that the models had not seen before (see also major comment 1 and major comment 2b).

28. L. 327-330: How can a cold bias in the Alpine region, or a negative mean sea-level pressure bias, cause the biases in the annual foehn occurrence? I suggest that these models must either be biased in the north-south *pressure gradients* during foehn, or have less synoptic conditions conducive to foehn formation, which would be counterintuitive if they reproduce the frequency of extratropical cyclones over Europe.

29. L. 341-344: As there are now two studies claiming an increase in the occurrence of south foehn in spring and a decrease in late summer/early autumn, it would be very interesting to hear from the authors some possible hypotheses as to why this shift might occur under future climate. Will there be a shift in the seasonality of extratropical cyclones passing over Europe? Is the propagation and breaking of Rossby waves expected to change? I would be interested to hear some speculation from the authors in this regard. This is a relevant question for future research, as the authors rightly point out on pp. 357-358.

30. L. 345ff: I see two more caveats of the study that should be mentioned here:
    a) The XGBoost models are not able to able to fully replicate the seasonality of foehn occurrence in the historical period (perhaps due to the lack of a split between the training and test datasets; see main comment 1).
    b) The statistical approach is unable to capture local effects on foehn occurrence, such as the effect of local cold-air pools. This makes training more difficult, and potential future changes in the occurrence of cold-air pools and the effects on foehn occurrence are also not captured by the approach. The use of climate models that explicitly resolve foehn flows could mitigate this limitation.

**Textual comments:**

1. L. 25: I think Greenland has just one ice sheet, i.e., the "Greenland Ice Sheet".
2. L. 26: I would be consistent and state the location for each impact, i.e., damaging rice crops *in Japan.*
3. L. 27: It should be "Baumann et al. (2001)". Also check the order of authors with the journal guidelines.
4. L. 31: Is "FAO" the author names?
5. L. 53: "conditions for foehn" sounds a bit strange to me. Maybe "conditions associated with foehn" is better suited?
6. L. 114: "…, which absolute values were expected to be due to conditions close to the free atmosphere" → "…, as absolute values of wind speed in the free atmosphere were expected to be captured realistically"
7. L. 70-71: "…, model weighting is likely to improve predicting a model mean and smaller uncertainties, …" → This sounds strange. Maybe rephrase to "…, model weighting is likely to improve the model mean and reduce uncertainties"?
8. L. 79: "producing south foehn" → "producing synoptic to mesoscale conditions associated with south foehn"
9. L. 94: "adjusting a NWP to observations" → "assimilating observations including, amongst others, radars, satellites and radiosondes into the integrated forecast system of ECMWF"
10. L. 268: "… in the historical period, summer across all regions …" → "… in the historical period across all regions…"
11. L. 293: "eastern Alps" → "Eastern Alps" (is capitalized in other occurrences)

**References:**

Ban, N., Caillaud, C., Coppola, E., Pichelli, E., Sobolowski, S., Adinolfi, M., Ahrens, B., Alias, A., Anders, I., Bastin, S., Belušić, D., Berthou, S., Brisson, E., Cardoso, R. M., Chan, S. C., Christensen, O. B., Fernández, J., Fita, L., Frisius, T., Gašparac, G., Giorgi, F., Goergen, K., Haugen, J. E., Hodnebrog, Ø., Kartsios, S., Katragkou, E., Kendon, E. J., Keuler, K., Lavin-Gullon, A., Lenderink, G., Leutwyler, D., Lorenz, T., Maraun, D., Mercogliano, P., Milovac, J., Panitz, H.-J., Raffa, M., Remedio, A. R., Schär, C., Soares, P. M. M., Srnec, L., Steensen, B. M., Stocchi, P., Tölle, M. H., Truhetz, H., Vergara-Temprado, J., de Vries, H., Warrach-Sagi, K., Wulfmeyer, V., and Zander, M. J.: The first multi-model ensemble of regional climate simulations at kilometer-scale resolution, part I: evaluation of precipitation, Clim. Dyn., 57, 275-302, https://doi.org/10.1007/s00382-021-05708-w, 2021.

Courvoisier, H. W., and Gutermann, T.: Zur praktischen Anwendung des Föhntests von Widmer, Arbeitsber. MeteoSchweiz, 21, 7 pp., https://www.meteoschweiz.admin.ch/dam/jcr:0d888977-6ba6-4651-8694-5583dd385488/arbeitsbericht21.pdf, 1971.

Drobinski, P., Steinacker, R., Richner, H., Baumann-Stanzer, K., Beffrey, G., Benech, B., Berger, H., Chimani, B., Dabas, A., Dorninger, M., Dürr, B., Flamant, C., Frioud, M., Furger, M., Gröhn, I., Gubser, S., Gutermann, T., Häberli, C., Häller-Scharnhost, E., Jaubert, G., Lothon, M., Mitev, V., Pechinger, U., Piringer, M., Ratheiser, M., Ruffieux, D., Seiz, G., Spatzierer, M., Tschannett, S., Vogt, S., Werner, R., and Zängl, G.: Föhn in the Rhine Valley during MAP: A review of its multiscale dynamics in complex valley geometry, Q. J. Roy. Meteor. Soc., 133, 897–916, https://doi.org/10.1002/qj.70, 2007.

Dürr, B.: Automatisiertes Verfahren zur Bestimmung von Föhn in Alpentälern, Arbeitsber. MeteoSchweiz, 223, 22 pp., https://www.meteoschweiz.admin.ch/dam/jcr:3ed2aec8-0901-417a-acc3-

8be11cce440a/Foehnindex_Arbeitsbericht_223_Automatisiertes_Verfahren_zur_Bestimmung_von_Foehn_in_Alpent aelern_de.pdf, 2008.

Gutermann, T., Dürr, B., Richner, H., and Bader, S.: Föhnklimatologie Altdorf: Die lange Reihe (1864-2008) und ihre Weiterführung, Vergleich mit anderen Stationen, Arbeitsber. MeteoSchweiz, 241, 53 pp., https://doi.org/10.3929/ethz-a-007583529, 2012.

Haid, M., Gohm, A., Umek, L., Ward, H. C., Muschinski, T., Lehner, L., and Rotach, M. W.: Foehn–cold pool interactions in the Inn Valley during PIANO IOP2, Q. J. Roy. Meteor. Soc., 146, 1232–1263, https://doi.org/10.1002/qj.3735, 2020.

Haid, M., Gohm, A., Umek, L., Ward, H. C., and Rotach, M. W.: Cold-air pool processes in the Inn Valley during föhn: A comparison of four cases during the PIANO campaign, Bound.-Lay. Meteorol., 182, 335–362, https://doi.org/10.1007/s10546-021-00663-9, 2022.

Jansing, L. and Sprenger, M.: Thermodynamics and airstreams of a south foehn event in different Alpine valleys, Q. J. Roy. Meteor. Soc., 148, 2063–2085, https://doi.org/10.1002/qj.4285, 2022.

Jansing, L., Papritz, L., Dürr, B., Gerstgrasser, D., and Sprenger, M.: Classification of Alpine south foehn based on 5 years of kilometre-scale analysis data, Weather Clim. Dynam., 3, 1113–1138, https://doi.org/10.5194/wcd-3-1113-2022, 2022.

Jansing, L., Papritz, L., and Sprenger, M.: A Lagrangian framework for detecting and characterizing the descent of foehn from Alpine to local scales, Weather Clim. Dynam., 5, 463-489, https://doi.org/10.5194/wcd-5-463-2024, 2024.

Mayr, G. J., Armi, L., Gohm, A., Zängl, G., Durran, D. R., Flamant, C., Gaberšek, S., Mobbs, S., Ross, A., and Weissmann, M.: Gap flows: Results from the Mesoscale Alpine Programme, Q. J. Roy. Meteor. Soc., 133, 881–896, https://doi.org/10.1002/qj.66, 2007.

Miltenberger, A. K., Reynolds, S., and Sprenger, M.: Revisiting the latent heating contribution to foehn warming: Lagrangian analysis of two foehn events over the Swiss Alps, Q. J. Roy. Meteor. Soc., 142, 2194–2204, https://doi.org/10.1002/qj.2816, 2016.

Mony, C., Jansing, L., and Sprenger, M.: Evaluating Foehn Occurrence in a Changing Climate Based on Reanalysis and Climate Model Data Using Machine Learning, Weather Forecast., 36, 2039–2055, https://doi.org/10.1175/WAF-D-21-0036.1, 2021.

Richner, H., Dürr, B., Gutermann, T, and Bader, S.: The use of automatic station data for continuing the long time series (1864 to 2008) of foehn in Altdorf. Meteorol. Z., 23, 159–166, https://doi.org/10.1127/0941-2948/2014/0528, 2014.

Saigger, M. and Gohm, A.: Is it north or west foehn? A Lagrangian analysis of Penetration and Interruption of Alpine Foehn intensive observation period 1 (PIANO IOP 1), Weather Clim. Dynam., 3, 279–303, https://doi.org/10.5194/wcd-3-279-2022, 2022.

Seibert, P., Feldmann, H., Neininger, B., Bäumle, M., and Trickl, T.: South foehn and ozone in the Eastern Alps – case study and climatological aspects, Atmos. Environ., 34, 1379–1394, https://doi.org/10.1016/S1352-2310(99)00439-2, 2000.

Tian, Y., Duarte, J. Q., and Schmidli, J.: A station-based evaluation of near-surface south foehn evolution in COSMO-1. Q. J. Roy. Meteor. Soc., 150, 290–317, https://doi.org/10.1002/qj.4597, 2024.

Umek, L., Gohm, A., Haid, M., Ward, H. C., and Rotach, M. W.: Large-eddy simulation of foehn–cold pool interactions in the Inn Valley during PIANO IOP 2, Q. J. Roy. Meteor. Soc., 147, 944–982, https://doi.org/10.1002/qj.3954, 2021.

Umek, L., Gohm, A., Haid, M., Ward, H. C., and Rotach, M. W.: Influence of grid resolution of large-eddy simulations on foehn-cold pool interaction, Q. J. Roy. Meteor. Soc., 148, 1840–1863, https://doi.org/10.1002/qj.4281, 2022.

Widmer, R.: Statistische Untersuchungen über den Föhn im Reusstal und Versuch einer objektiven Föhnprognose für die Station Altdorf. Vierteljahresschr. Naturforsch. Ges. Zürich, 111, 331–375, https://www.ngzh.ch/archiv/1966_111/111_3-4/111_20.pdf, 1966.

Würsch, M. and Sprenger, M.: Swiss and Austrian foehn revisited: A Lagrangian-based analysis, Meteorol. Z., 24, 225–242, https://doi.org/10.3929/ethz-b-000103254, 2015.

---

## Referee Comment (RC2)

**Review egusphere-2024-670**

Analysing CMIP5 EURO-CORDEX models in their ability to produce south foehn and the resulting climate change impact on frequency and spatial extent over western Austria (Maier et al.)

**Summary**

The manuscript studies south foehn in parts of the Austrian Alps and how its spatio-temporal characteristics might change in the future. It uses an algorithm to diagnose foehn at several locations. This foehn diagnosis is then used as a response variable in the training of machine learning models to diagnose foehn using reanalysis and climate simulation data, respectively. At first glance the paper seems to be a replica of Mony et al. (2021) albeit with spatially higher resolution reanalysis and climate simulation data and for the Alpine region immediately east. At a closer look, though, the paper has major shortcomings and lacks reproducibility in essential details as explained in the next section. Therefore rejection of the paper is recommended.

**Reasons for rejection**

1. **Misclassification of foehn events:** The objective foehn classification method was inappropriately applied by extending a potential temperature difference specific to one location to all locations. Foehn strongly depends on the topography such as the presence of tributary valleys upstream from which cold air can drain into the valley where the foehn blows, the presence of cold pools, the distance from the crest, and the altitude difference to the upstream crest. However, the authors use the potential temperature difference between crest and valley station from Fig. 4a in Plavcan et al (2014), which was determined for a specific location (Ellboegen). This threshold needs to be derived from the data for each station separately. Since the remainder of the paper builds on the foehn classification its results are dubious.

2. **Inappropriate clustering of foehn locations:** Foehn frequencies will only to a small degree be affected by being somewhat more west or east on the northern side of the Alpine crest and to a much larger degree by their specific location and the specific processes at work (cold pool formation, flow separation, drainage of cold air from tributary valleys, …). The locations aggregated into the three groups are thus expected to have foehn frequencies that might differ by one order of magnitude. All the analyses starting with the foehn classification would need to be performed separately for each location and only at the very end an examination of regional differences be performed.

**Major comments**

1. Since foehn is very sensitive to the details of the underlying topography and one of the major goals of the paper is to investigate future changes of foehn frequency, ERA5 reanalysis data should be interpolated to the finer grid of the climate simulations (instead of the other way round as done in the paper).

2. Unclear how a foehn *day* is diagnosed from hourly measurements. The paper states "whereas the most extended event observed within the day's 24 hours is used". Does this mean that a "foehn day" is one with at least one hour of foehn?
3. Foehn diagnosis not reproducible and possibly inappropriate:
    a. Table 2 does not state a wind sector for foehn at a valley station. Please supply. If none was used but only the exceedance of 50 (90) % of wind speeds at the crest then e.g. situations with southerly flow aloft but diurnal upvalley winds will be misclassified.
    b. Wind speed thresholds at the crest station (given that the wind is in the appropriate sector) are unnecessary and eliminate many foehn hours. Potential temperature difference and appropriate wind at the valley station will suffice. No wonder that hardly any widespread events are classified if they require top 10% wind speeds at crest.
    c. Missing results of the foehn diagnosis
4. In-sample model results: From what I read in the manuscript, the authors performed a hyperparameter grid search to determine the tuning parameters of the xgBoost model. The data used for the grid search can then no longer be used to determine the expected model performance.

Minor comments
1. The title is too long and confusing as it implies that south foehn leads to climate change
2. Line 72: Does the statement of the "binary nature" of foehn contradict how it is treated in 3 categories in equ(1)?
3. Note that you do not need pressure data to use the foehn classification method of Vergeiner (2004). As this method checks whether air could have dry-adiabatically descended from the crest station to the valley station, reducing the temperature from the crest station to the valley station with a (linear) dry-adiabatic temperature gradient will do and only needs station elevations.
4. Table 2: Is the crest wind speed threshold chosen only from the wind speed distribution of winds from 135 - 225 degrees or from all directions (which would not be appropriate)?

---

## Author Comment (AC1)

**Author Response for Manuscript: Analysing CMIP5 EURO-CORDEX models in their ability to produce south foehn and the resulting climate change impact on frequency and spatial extent over western Austria**

Philipp Maier[1], Fabian Lehner[1], Tatiana Klisho[1], and Herbert Formayer[1]

[1]Institute of Meteorology and Climatology, University of Natural Resources and Life Sciences, Vienna, Austria

**Correspondence:** Philipp Maier (philipp.maier@boku.ac.at)

**General Remarks**

We first want to thank the editor and referees for taking the time to judge our work and contribute in its growth. Every comment is greatly appreciated and will greatly improve the manuscript. The reviews helped us in noticing shortcomings in our current revision and we are positive, that we can address all comments in a satisfying way to make our manuscript fit for publication within a few weeks.

As we will point out in the following section addressing the rejection by referee 2, we take the concerns regarding foehn classification and station clustering very seriously. We did not clarify, that we are interested in a regional perspective on foehn and not focusing on individual valleys. Therefore, a lot of our decisions made during handling the weather stations seem counter-intuitive. This will be edited in the revised manuscript. We kindly ask for a reconsideration by referee 2 before the final decision because of the following reasons, which are also explained in more detail in the coming sections:

- We are positive, that we can address the simplifications and deviations we conducted in classifying foehn using OFC (Vergeiner, 2004; Drechsel and Mayr, 2008). These simplifications are clarified mainly in our answers to the first rejection reason and the major comments of referee 2.

- We did not put enough emphasis on the performance validation and display of the training data derived with OFC in the methods section as well as in the results and discussions. We apologise for that and will edit that in the newer version of the manuscript. We are in contact with the 'Alpine Research Group Foehn Rhine Valley/Lake Constance (AGF)' and the Austrian weather service Geosphere Austria to get several stations with daily foehn classification based on the method of Dürr (2008) for our target area. This allows an objective comparison of our simplified approach at these locations and will be included in the manuscript. With this, we hope to answer the first rejection reason of referee 2 and the major comments of referee 1.

- We plan to base our sub-region definition on a clustering technique to identify valley stations with similar historic foehn characteristic. With that approach and more explanation why we require to group individual stations, we strive to solve the second rejection reason of referee 2.

– The machine learning training was already done with a subset of training and evaluation data, which we failed to communicate in the current revision. By adding (more) performance parameters based on the evaluation data set and explaining the training process in more detail, we are positive, that the major comments by referee 1 and referee 2 can be addressed satisfyingly.

– As we judge by the small number of comments regarding the assessment of EURO-CORDEX model performance, this part of the manuscript was received well. We therefore judge the latter half our study to be scientifically sound and the trend analysis to be valid within its limitations.

In the following sections, we answer all comments carefully in more detail. For better readability, we included the original comments in a gray font. We are open for further fruitful discussion and are looking forward to the decision by the editor, if re-submission is encouraged.

Sincerly,

Philipp Maier
(on behalf of the author team)

**Response to Referee 1 (https://doi.org/10.5194/egusphere-2024-670-RC1)**

**Synthesis**

The present study addresses a relevant topic, namely potential changes in the occurrence of foehn events under a changing climate, thereby focusing on western Austria in the Alps. Following a methodology similar to that of a previous study, the authors employ six XGBoost models to indirectly address this question. These models are trained using ERA5 variables in order to predict the occurrence of both localized and widespread foehn events across three subregions, with station-based foehn detection data providing the training labels. Subsequently, the machine-learning models are applied to CMIP5-CORDEX climate simulations for present-day and future climate scenarios. The authors find a shift in the seasonality of foehn events under future climate, with foehn becoming more frequent in spring and less frequent in autumn.

The manuscript tackles a relevant research gap and chooses a suitable methodological approach. The study would therefore be a valuable addition to the existing foehn literature, particularly given the limited understanding of how the frequency of foehn events may be influenced by global warming. However, there is a need for a more detailed explanation of the training and validation process of the XGBoost models, as highlighted in the first major comment. Additionally, there is room for refinement in both the structure and content of the manuscript: currently, the methods section is lengthy in contrast to the relatively concise and descriptive results section. Expanding the discussion of the results, as suggested in the second major comment, would significantly enhance the manuscript. Furthermore, there are many minor comments to be addressed that would further improve readability, consistency, and the overall quality of the manuscript. For these reasons, I recommend that the authors address these concerns if the manuscript is to be considered for publication in WCD.

We consider ourselves privileged and are very thankful to receive such a detailed and precise review, which will significantly improve the manuscript. The synthesis values our work and we are delighted, that the manuscript is considered relevant, although the structure and textual distribution of the manuscript needs refinement. We first want to acknowledge the textual comments and literature suggestions, which will enhance readability and information density of our study.

**Major Comments**

1. **Approach to training and validation**: According to the reviewers' understanding, the authors opted not to partition the ERA5 and the OFC data into distinct training and test sets. Typically, machine learning models are trained on a subset of the data, while another subset remains unseen for the model and is reserved for testing and performance evaluation (see, e.g., Mony et al., 2021). Consequently, comparing performance metrics with other studies might be somewhat misleading, as the evaluation relies on the time period used for model training. I understand that the authors conduct an indirect performance evaluation by comparing the seasonal foehn frequency derived from OFC data in the time period of 2011-2021 to that of ERA5 and EURO-CORDEX data derived from the period 1991-2020. Nevertheless, I am curious

how the authors intend to demonstrate that the current approach does not lead to overfitting and consequently render the results less generalizable, particularly considering the differences we see between OFC and ERA5, but especially between OFC and EURO-CORDEX models in Figure 5. The temporal mismatch of the time period where the OFC data is available (2011-2021) compared to the historical time period used for indirect validation (1991-2020) makes it even more challenging to judge whether the XGBoost models have been successfully trained and applied to the present-day EURO-CORDEX simulations. Please comment on these aspects and discuss them in the manuscript.

The division of the training data in a subset used for training and evaluation is common practice and we want to apologise for not mentioning that we indeed performed that. The training data was split into a training subset containing 80 (90) % of the data and into an evaluation subset containing 20 (10) % for the training of the model for localised (wideseread) foehn events. The training subset was then used for the hyperparameter grid search with five folds, culminating in the hyperpameters with the best overall accuracy. Then, the model with the best hyperparameters is fitted to the training subset and the evaluation subset is used for optimising this process. Consequently, the inclusion of the evaluation data set, not yet seen by the XGBoost model is counteracting over fitting. The performance parameters like the accuracy were then calculated using the full training data again, which is misleading will be displayed based on the evaluation subset for better comparison to Mony et al. (2021). The split of the training data into subsets of actual training and evaluation was not communicated in the manuscript and will be added, as this was an inaccuracy. We further will include the ERA5 predictions from 2011-2021 to clarify the indirect performance validation in Figure 5. We hope, with those two additions, we can overcome the shortcoming mentioned in this comment.

2. The authors have made a clear distinction between the descriptive results and their interpretation, which is a valid option. However, I believe that**a more extensive discussion of the results is necessary, extending beyond what is currently presented in the conclusions.** Whether this extended discussion is incorporated at the end of each individual results section, integrated within more comprehensive conclusions, or presented as a distinct discussion section is up to the authors. However, in my opinion, such an addition would create a more equal balance between the actual results and discussion within the paper, especially compared to the current methods section. I give several suggestions for additional interpretation and discussion of the results below (further suggestions are provided in the minor comments as well):

We agree that an extension of results is required for a more justified balance between methods and results. By adding the following valuable suggestions, we will naturally extend the discussion section to a suitable proportion of the whole manuscript.

    a) **Interpret and discuss the seasonal foehn occurrence in the three regions for the present-day period according to OFC**: Figure 3 presents intriguing results and could be discussed in more detail. Is this the first "climatology" of foehn occurrence in Vorarlberg and Tiroler Oberland? The authors could highlight this more explicitly if no such prior publication exists. Additionally, several open questions arise when looking at the differences in the seasonal cycle of foehn between the three regions: Why does Vorarlberg feature substantially less foehn days during most months compared to the Tirol regions? Why do widespread events appear to be more prevalent in Tiroler Unterland compared to the other regions? Considering the size of the study area, one would anticipate that foehn events occur under similar synoptic weather conditions. The authors could validate this assumption by computing ERA5 composites for the mean synoptic situation during foehn days in 2011-2021 across the three regions.

Local effects, rather than synoptic differences, are probably responsible for the above-mentioned differences between the three regions. The authors mention the north-south valley axis of the Wipp Valley as a factor favoring foehn breakthrough (L. 38 and L. 302). Are none of the other stations located in a valley that is also north-south oriented? It is worth noting that several other factors influence the foehn climatology at specific stations. Foehn is more likely to occur at stations that are closer to the main Alpine crest, both horizontally and vertically (e.g., Gutermann et al., 2012). Additionally, a deep incision in the crest, such as the Brenner gap, favors foehn penetration, especially during shallow foehn episodes that are primarily driven by the overflow of colder air from the southern to the northern side (e.g., Jansing et al., 2022). Another crucial factor to consider is a station's tendency to cold-air pooling. During the colder months, the foehn flow at lower-elevation stations in valleys often fails to reach the ground due to the stable valley air mass hindering its breakthrough (e.g., Drobinski et al., 2007; Haid et al., 2022). How strong is this effect at the different stations across the three regions? I am particularly surprised by the higher occurrence of widespread events in the Tiroler Unterland compared to other regions, as I had previously assumed that stations in the Inn Valley (such as Innsbruck, Jenbach, and Kufstein) are prone to cold-air pooling, especially in autumn and winter. Moreover, it is worth noting that local cold-air pools can pose challenges for the accurate prediction of foehn occurrence by the XGBoost models, as foehn-prone conditions might not lead to foehn breakthrough under such circumstances. I encourage a discussion of these factors in relation to the seasonal foehn occurrence diagnosed from OFC data, as well as their implications for the XGBoost predictions.

Figure 3 is indeed the first climatology for a combination of stations for these three regions. So far, climatologies are provided for individual stations, particularly by the 'Alpine Research Group Foehn Rhine Valley/Lake Constance (AGF)' in Vorarlberg and for the stations close to Innsbruck in the works of Vergeiner (2004), Drechsel and Mayr (2008) or Plavcan et al. (2014), but no combination was found in our literature research. As referee 2 indicated, the clustering of stations also bears other challenges, which we will address in the next revision of the manuscript. We attribute the different frequency of foehn in Vorarlberg and the Tirol areas to the different flow characteristics for the Rhine and Wipp valley, as stated by Zängl et al. (2004). The high frequency of widespread events was attributed to the close-to-ideal orientation of the Wipp valley. But as you correctly pointed out, the Wipp-valley is not the only valley with north-south orientation and therefore, this cannot be the only reason for widespread events. We will also include the deep incision of the Brenner gap and the subsequent air splitting in both directions of the Inn valley, affecting multiple stations during one event as explanation, amongst others. Your comment as well as the comments of referee 2 showed us, that we have to elaborate more on the local effects of foehn occurrence.

Due to the smoothed topography in complex terrain, ERA5 and the EURO-CORDEX data are both not able to produce cold air pools in valleys of the studied size, which is one of the reasons why we chose the selected approach of linking synoptic patterns with station data. We believe, that we will gain additional information about stations prone to cold air-pooling in the comparison of the region-wide algorithm and individual stations we suggested in the response to referee 2 and will interpret the results accordingly. We are thankful for the literature suggestions in this regard and will put emphasis in the conclusion, that our method is only suitable for foehn events likely to fully break through into valleys, where no cold air pool is present. We also will further analyse the synoptic conditions of the different regions and intensities and point out differences and similarities as suggested and include it in the manuscript or the appendix.

b) **Provide a more comprehensive description and interpretation of the training results**: The authors might consider incorporating metrics such as the correct alarm ratio and probability of detection (as suggested in minor comment 16), as the accuracy metric may present an overly optimistic view on the model performance given the imbalanced dataset. This would enable a more precise evaluation of the performance discrepancies among the XGBoost models. For instance, the correct alarm ratio for Vorarlberg loc. / wide., as well as the probability of detection and the correct alarm ratio for Tiroler Unterland loc. / wide. seem to be lower compared to other metrics. Moreover, the authors could consider showing the importance of additional features, or at least indicate whether they hold nearly equal importance or are considerably less significant compared to the most important feature (Table 4). Additionally, Figure 5 could be expanded to multiple panels to illustrate the results for all three regions and both categories (localized/widespread). This represents one of the main results of the study. Presently, with the existing Figure 5, readers are unable to discern how ERA5, the weighted, and the unweighted EURO-CORDEX models capture the seasonality across different regions and categories. The manuscript would benefit significantly from including these results and discussing them within the text.

We will add the correct alarm ratio and probability of detection based on the evaluation subset and discuss it properly for more precise validation of the XGBoost training, also in comparison to Mony et al. (2021). We further will provide an additional Figure besides Table 4 to illustrate feature importance distribution and split Figure 5 into multiple panels, as this is more precise in communicating our main results. In consideration of the comments of both referees, we realised that we did not put enough emphasis on the earlier processes of our study and focused too strong on the EURO-COORDEX analysis. This will be changed in the new version of the manuscript.

c) **Revisit the description and interpretation of the XGBoost models applied to ERA5 and the EURO-CORDEX models for the historical period**: Unfortunately, there is a temporal discrepancy between the time periods depicted for the training data (2011-2021) based on OFC and the predictions from the XGBoost models (1991-2020). How does the ERA5 seasonality look like when only considering the training period from 2011-2021? The same question is also valid for the XGBoost predictions using EURO-CORDEX data. This comment is also related to comment

a) further above.

We also would have preferred a longer time period for training, but were unfortunately limited by the combined data availability of all stations. As discussed in point a) and b), we will include the ERA5 results for the training period 2011-2021 in Figure 5 to bridge the gap between the 30-year climate normal period, which is a better representation for the EURO-CORDEX models as individual days and years are not matching observations in the historical period, and split Figure 5 into multiple panels. We will note this shortcoming in the limitations.

**Minor Comments**

1. The title ("... produce south foehn ...") is somewhat misleading in the sense that the EURO-CORDEX models do not actually simulate foehn flows that resemble actual foehn flow characteristics, since their resolution, as the authors correctly state in their manuscript, is too coarse to resolve individual foehn valleys. These models only represent the synoptic (and mesoscale) conditions that are typically associated with the occurrence of foehn flows in the Alps. Consider rephrasing the title of the manuscript.

   Acknowledged. As we also have the impression, that we emphasised the model performance too heavily, our new preposition for the title (including suggestion from referee 2) is: "Analysing the future trends of synoptic patterns enabling south foehn over western Austria in CMIP5 EURO-CORDEX models"

2. L. 9: Similar to the first minor comment, the EURO-CORDEX models do not "produce south foehn" in the sense that these flows are resolved in the models. Instead, foehn is diagnosed using the atmospheric fields that represent the typical synoptic to mesoscale weather situations during foehn and non-foehn conditions.

   It will be clarified wherever it is mentioned in the running text, that the EURO-CORDEX models do not produce south-foehn but just produce atmospheric fields, which are present during foehn conditions.

3. L. 36ff: Please consider expanding this sentence. It is, in my opinion, oversimplifying the foehn research in the Alpine region, which is not limited to two field campaigns. If you are referring to the MAP field campaign, it would be more accurate to cite Mayr et al. (2007) and Drobinksi et al. (2007) to adequately represent the MAP findings concerning foehn flows. Furthermore, there has been a recent, more local-scale campaign focusing on foehn-cold-air-pool interactions (Haid et al., 2020, 2022.). Moreover, there exists a range of recent publications that focus on the representation of Alpine foehn flows in mesoscale to large-eddy simulations, aiming to understand the physical processes governing foehn flows and to improve the quality of foehn simulations (e.g., Seibert et al., 2000; Würsch and Sprenger, 2015; Miltenberger et al., 2016; Umek et al., 2021; Jansing and Sprenger, 2022; Jansing et al., 2022; Saigger and Gohm, 2022; Umek et al., 2022; Jansing et al., 2024; Tian et al., 2024). Consider incorporating this important aspect of foehn research in an additional statement and referring to selected publications.

The history of foehn research will be expanded beyond the stated milestones, including the interaction of foehn with cold air pools and large-eddy simulations. We want to apologise for overseeing this part of research and are thankful for literature suggestions.

4. L. 40: The descent of foehn in the framework of hydraulic theory does not occur due to the flow transition into critical or supercritical flow, but due to cross-Alpine density differences (buoyancy-driven, see, e.g., Mayr et al., 2007).

   The paragraph on hydraulic foehn theory will be clarified.

5. L. 41ff: Forecasting foehn using statistical methods extends further back in time. Widmer (1966) and Courvoisier and Gutermann (1971) already developed an index to predict foehn at Altdorf, Switzerland. Moreover, Dürr (2008) also developed an objective foehn classification method using station data.

   The paragraph on forecasting foehn will be extended with the mentioned earlier works. It is a grievous oversight, that we did not include the study of Dürr (2008) in this paragraph. As of now, he have acquired station data derived from this method to verify our approach at some stations in our study region, gratefully provided by the 'Alpine Research Group Foehn Rhine Valley/Lake Constance (AGF)'. Therefore, more emphasis will be put on that method of foehn detection as well.

6. L. 46: This is actually not true, there are by now regional climate projections with a spatial resolution down to 2.2 km (e.g., Ban et al., 2021). In these models, one could actually at least partially resolve the foehn flows and thus explicitly investigate foehn flows under future climate.

   The works of Ban et al. (2021) will be cited, and the addition 'for future scenarios' will be added to L46.

7. L. 51: Given the similarity in approach, it would be beneficial to summarize the main findings of Mony et al. (2021) in one or two sentences. This would sharpen the research addressed in the study, specifically the investigation of future foehn occurrence over Austria, a topic that has not been explored previously. For instance, it has remained uncertain whether the conclusions drawn by Mony et al. (2021) could be replicated with other climate models and applied to foehn regions in the Eastern Alps.

   The main findings of Mony et al. (2021) will be summarised and their training performance metrics will be compared to ours in the new version of the manuscript. Additionally, we will refine own research question.

8. L. 59-73: A substantial paragraph of the introduction is devoted to a discussion of the advantages and disadvantages of ensemble weighting. I wonder if at least part of this discussion would be better placed in the methods section. The current structure of this paragraph overemphasizes this particular method and focuses too little on the actual aims of the study. Therefore, the author may consider restructuring this part of the introduction to better align with the overall focus of the research.

   The whole manuscript puts a strong emphasis on model weighting while not providing enough room for station handling. This will be streamlined to a better proportion or moved to the appendix.

9. Fig. 2: Nice flowchart! Why not drawing an arrow from "11 years of daily training data" to "Training of two XGBoost models per region" as well?

Thank you! Arrow will be added in the flowchart in Figure 2.

10. Table 1: How is the model main ridge defined? Please specify this.

The main ridge is defined by locating the pixel with the highest elevation for every latitude in the rectangular grid over the Alpine area. This will be specified.

11. Table 1 vs. L. 97-99: Table 1 is not fully consistent with the text; for instance, the cross-Alpine potential temperature gradient is not mentioned in the text. Additionally, could the authors provide further insight into their rationale behind selecting the cross-Alpine pressure gradient within $\pm$ 1° latitude relative to the model's main ridge as a feature (only 37 grid points), compared to the inclusion of every grid point for the geopotential difference on 500 hPa (925 grid points)? The same question applies to the cross-Alpine potential temperature gradient. Moreover, the authors might consider elaborating on why they opted for the relative humidity difference with respect to the monthly mean as an additional feature, especially since it is the only variable where an anomaly is considered. It would be beneficial for readers to gain a deeper understanding of the motivation behind the selection of the feature matrix input.

More emphasis will be put on the motivation behind the selection of the features and the pre-processing based on our physical understanding of the foehn process.

12. L. 100-101: This is actually not the study domain (not the red box in Fig. 1a), but the domain considered to extract features from ERA5.

Thankfully noted.

13. L. 110: Can the authors specify how many models are included in this OEKS15 selection?

Will be specified.

14. L. 117ff: In my opinion, this paragraph could be shifted to the beginning to the next paragraph, as it is directly related to the OFC used to generate the training labels. If done so, I would also suggest to rename section 2.1, e.g., to "2.1 Reanalysis data and climate projections".

Thankfully noted.

15. L. 141-142: Judging from Fig. 4b, Vorarlberg does not feature a less dense network of stations, but simply is the smaller region.

Argumentation will be changed.

16. L. 179-182: As the authors correctly point out, the accuracy (acc) is not a suitable metric for a highly unbalanced dataset like the one presented here. They could therefore consider to also evaluate the correct alarm ratio (TP / (TP + FA)) and

the probability of detection (TP / (TP + ME)), which would give the reader a more comprehensive picture of the model performance on the training dataset.

As mentioned in the comment regarding major comment 2b), this will be changed.

17. L. 199ff / Eqs. 5-7: As the authors compare two different time periods (1991-2020 vs. 2011-2021), this correction is only valid if the annual frequency, the seasonality and the inter-annual variability in the period 1991-2010 do not differ too much from the period 2011-2021. Unfortunately, this cannot be tested with the training data used in this study. It should be noted that years with an unusually high or low foehn frequency often co-occur over several years (e.g., Richner et al., 2014). Did the authors try a correction where they calculated the biases using only the period 2011-2020 (i.e., N = 10 instead of N = 30)?

As mentioned in the response to major comment 2c), we will include the ERA5 seasonality of the training period (2011-2021) to better discuss the difference between the eleven-year-long period and the 30-year-long climate period in Figure 5, which will be addressing the different time periods implicitly. Further, we will calculate the biases for the training period only and include them as first entry above the full ERA5 period in Figure 4. The general reason why we decided on calculating the biases of the EURO-CORDEX models against the training period was, that ERA5 itself struggles to capture the seasonality. By calculating biases against ERA5, we would implicitly rank the ability of EURO-CORDEX models to capture foehn seasonality based on an already biased baseline. Therefore, the training data was used.

18. L. 205ff: Why is the seasonality bias ($b_{m,seasonality}$) either added or subtracted from the monthly foehn events before computing the standard deviation?

The seasonality bias is added (subtracted) to the months, which show less (more) foehn occurrence compared to the training data. This is done to ensure that after subtracting the annual bias, only the seasonality is changed and not the absolute numbers of annual foehn events, which are then used for calculating the inter-annual variability. In deeper analysis, we realised that this is actually not necessary as the effects of the bias-adjustment from the seasonality cancel out when summing over a whole year again in Equation 7. It is therefore more useful to subtract the annual bias $b_{m,annual}$ when calculating the bias for the inter-annual variability $b_{m,variability}$ in Equation 7, which will be adjusted in the revised manuscript.

19. Section 2.4: This section is rather lengthy, describing the methodological procedure of weighting of the different simulations. It therefore appears imbalanced, especially compared to the relatively short results section. The authors might consider shifting part of this explanation into an appendix (e.g., appendix D).

Part of this explanation of the weighting process will be shifted to the appendix to enhance readability and flow of the whole paper.

20. L. 235, L. 248, L. 256, L. 274: The section titles reflect the methodology used rather than the results they present. Why not rename them, for example, to "3.1 Present-day foehn climatology based on station data and OFC", "3.2 Model

performance for present-day climate" (merge 3.2 and 3.3 as 3.2 is very short anyway), "3.3 Foehn occurrence under future climate"? These are just suggestions, but consider renaming the sections to make them more intuitive.

More elaborate and intuitive section titles will be selected.

21. Figure 3: The caption explains that the uncertainty bars show the standard deviation over the 11 years. In my opinion, it would be more intuitive to adjust the ranges to the minimum and maximum, or to the 10th and 90th percentiles. The current visualization is misleading as there are certainly not less than 0 foehn days in a given month.

We will consider shifting the uncertainty bars to the maximum and minimum for Figure 3 and further plots. The 10th and 90th percentile unfortunately are poor indicators for the eleven year long training data, as these would be located somewhere in between the values of individual years.

22. Figure 3: I suggest to show the average localized foehn days in a third panel and the temporal evolution (i.e., the inter-annual variability) in a fourth panel. This would allow the reader to get a more complete picture of the contrasting behavior of the three regions (e.g., Tiroler Unterland has less localized but more widespread events than Tiroler Oberland – really surprising to me!)

More panels including the localised foehn events and the inter-annual variability will be added to Figure 3.

23. L. 239-242: The text is not consistent with the figure. The figure shows that Tiroler Unterland has the highest overall annual foehn frequency and a secondary maximum in autumn due to widespread events. In the text, however, the authors claim that the Tiroler Oberland has the highest annual foehn frequency (17.9 %) and the highest number of widespread events. Which is correct?

This is indeed a textual mistake on our side. Tiroler Oberland yields the highest foehn frequency overall and the highest number of widespread events. We apologise for the confusion.

24. L. 281 and L. 284: Please be careful when talking about "significant trends", especially in the context of statistical trend analysis. Did the authors test the trends for significance using a statistical method?

In this context, 'significant trend' was used as a synonym for 'noteable trend'. It was not intended to imply, that statistical testing was done. However, we will conduct null-hypothesis significance testing in the revised manuscript.

25. Figure 7: Given the different trends for localized and widespread events in Figure 6, it would be interesting to see Figure 7 separately for localized and widespread events. If the authors feel that this is not appropriate in the main text, they could include it in the appendix (or a supplement).

A seasonality trend for widespread foehn events will be added in the main text or the appendix. We will keep a similar structure as Figure 6, providing one plot for both foehn events combined, and one just for widespread events.

26. L. 298-303: Here the authors could discuss the other possible explanations for the regional differences (see major comment 2a).

We will discuss further regional differences, as indicated by responses to the major comments and to referee 2.

27. L. 307-308: Consider comparing the correct alarm ratio and the probability of detection with Mony et al. (2021). Note that the comparison is somewhat misleading, as the authors did not apply the XGBoost models to a test dataset that the models had not seen before (see also major comment 1 and major comment 2b).

We will compare our results to Mony et al. (2021) more carefully, and clarify our metrics as mentioned in the response for major comment 1.

28. L. 327-330: How can a cold bias in the Alpine region, or a negative mean sea-level pressure bias, cause the biases in the annual foehn occurrence? I suggest that these models must either be biased in the north-south pressure gradients during foehn, or have less synoptic conditions conducive to foehn formation, which would be counterintuitive if they reproduce the frequency of extratropical cyclones over Europe.

This is a question we asked ourselves and as this research gap persists, we plan to investigate that in future studies. What we tried to state with the lines 327-330 is, that we know that there are biases existing in the Alpine regions. How these are linked to foehn has yet to be investigated.

29. L. 341-344: As there are now two studies claiming an increase in the occurrence of south foehn in spring and a decrease in late summer/early autumn, it would be very interesting to hear from the authors some possible hypotheses as to why this shift might occur under future climate. Will there be a shift in the seasonality of extratropical cyclones passing over Europe? Is the propagation and breaking of Rossby waves expected to change? I would be interested to hear some speculation from the authors in this regard. This is a relevant question for future research, as the authors rightly point out on pp. 357-358.

We will address some of the hypotheses we have come up with in the section for future research as your comment has inspired us, that it could potentially spark future research.

30. L. 345ff: I see two more caveats of the study that should be mentioned here:

   a) The XGBoost models are not able to able to fully replicate the seasonality of foehn occurrence in the historical period (perhaps due to the lack of a split between the training and test datasets; see main comment 1).

   b) The statistical approach is unable to capture local effects on foehn occurrence, such as the effect of local cold-air pools. This makes training more difficult, and potential future changes in the occurrence of cold-air pools and the effects on foehn occurrence are also not captured by the approach. The use of climate models that explicitly resolve foehn flows could mitigate this limitation.

We will add both remarks to the limitations.

**Response to the Referee 2 (https://doi.org/10.5194/egusphere-2024-670-RC2)**

**Summary**

The manuscript studies south foehn in parts of the Austrian Alps and how its spatio-temporal characteristics might change in the future. It uses an algorithm to diagnose foehn at several locations. This foehn diagnosis is then used as a response variable in the training of machine learning models to diagnose foehn using reanalysis and climate simulation data, respectively. At first glance the paper seems to be a replica of Mony et al. (2021) albeit with spatially higher resolution reanalysis and climate simulation data and for the Alpine region immediately east. At a closer look, though, the paper has major shortcomings and lacks reproducibility in essential details as explained in the next section. Therefore rejection of the paper is recommended.

We want to thank referee 2 for their review. We indeed used a similar machine learning approach like Mony et al. (2021) as one of many methodological approaches used in this study. We naturally also gave credit to that in the manuscript, but we also want to state, that our study is not a replica of Mony et al. (2021) as the analysis and weighting of EURO-CORDEX models for foehn as well as the training data itself are novel approaches. Further, although some clarification in formulation is required to improve readability and conciseness, we think that the results are indeed reproducible and hope to be able to convince you of that with the following responses.

**Reasons for Rejection**

We argue that the reasons for rejection can be solved within a major revision as follows:

1. **Misclassification of foehn events**: The objective foehn classification method was inappropriately applied by extending a potential temperature difference specific to one location to all locations. Foehn strongly depends on the topography such as the presence of tributary valleys upstream from which cold air can drain into the valley where the foehn blows, the presence of cold pools, the distance from the crest, and the altitude difference to the upstream crest. However, the authors use the potential temperature difference between crest and valley station from Fig. 4a in Plavcan et al. (2014), which was determined for a specific location (Ellboegen). This threshold needs to be derived from the data for each station separately. Since the remainder of the paper builds on the foehn classification its results are dubious.

   We consider foehn as a synoptic scale phenomenon with the locally different characteristics you mentioned in individual valleys. Still, the process of air aloft descending into valleys is similar in every location. Small potential temperature differences indicate, that air masses at the crest and in the valley location are similar and therefore a descent quite possibly occurred. This holds for any valley station. Therefore, the same thresholds as Plavcan et al. (2014) used for Ellboegen were used for other stations. The wind direction at the crest additionally ensures foehn-enabling conditions. While we agree that misclassifications in individual valleys at individual foehn events will occur with this method due to the potential temperature criteria being too restrictive or liberal, the goal of identifying a foehn event in the three sub-regions of western Austria (and not at individual stations) on a daily basis should be possible with sufficient accuracy. We

further have acquired classification data generated with the method of Dürr (2008), gratefully provided by the 'Alpine Research Group Foehn Rhine Valley/Lake Constance (AGF)' for several stations and will verify our classifications with them in the revised manuscript. If this verification yields positive results, the trend for future climate can still be fulfilled as such events would also be misclassified in the future as well.

2. **Inappropriate clustering of foehn locations**: Foehn frequencies will only to a small degree be affected by being somewhat more west or east on the northern side of the Alpine crest and to a much larger degree by their specific location and the specific processes at work (cold pool formation, flow separation, drainage of cold air from tributary valleys, . . . ). The locations aggregated into the three groups are thus expected to have foehn frequencies that might differ by one order of magnitude. All the analyses starting with the foehn classification would need to be performed separately for each location and only at the very end an examination of regional differences be performed.

Clustering of stations at the very end is not used because we wanted to highlight future foehn trends on a regional scale, independent of individual stations and valleys. On the one hand, we know, that our classification approach is not able to accurately classify every foehn event correctly at specific locations as the foehn frequencies differ. On the other hand, we think the regional approach would be more useful for the general public. Especially the forestry and agricultural sector in those regions can use the future foehn conditions to decide on which climate-resilient crops and trees to plant in the region as a whole and many other impact studies like e.g. on human health are also interested on a more regional scale. As typical synoptic patterns have to be persistent for foehn to form and these patterns can be prone to a trend, we therefore saw it as justified to group certain stations for different regions. By keeping in mind the special role of the Wipp valley and Rhine valley, we decided on the three regions you see in the manuscript. With the assumption of stronger cross-alpine pressure gradients yielding to more widespread foehn events and stronger wind speeds at the crest, we additionally decided to consider two classes of foehn. However this is of course a simplification as every location is different. For a better understanding of how individual stations perform and with the hope to mitigate the concerns for the clustering method, we will:

- Perform a clustering method (like k-means or a correlation matrix) after applying the foehn classification on individual valley stations. With this method, stations with the same behaviour and not only similar geographical location are clustered together. We assume, that this will change the regions only slightly but provides a justified basis for analysis.

- Show a heat map for the training period 2011-2021, indicating differences in foehn between individual stations and the regional machine learning algorithm classification.

**Major Comments**

1. Since foehn is very sensitive to the details of the underlying topography and one of the major goals of the paper is to investigate future changes of foehn frequency, ERA5 reanalysis data should be interpolated to the finer grid of the climate simulations (instead of the other way round as done in the paper).

The climate projections have a spatial grid resolution of around 11 km, whereas ERA5 has a resolution of around 30 km. Additionally their is a broad discussion on the "true" resolution of climate models. Typically it is assumed that it is a manifold (e.g. 3 times) of the grid resolution. While foehn is sensible to details of the underlying topography, these details in the Alpine region can not be resolved, neither in the ERA5 resolution nor in the finer EURO-CORDEX resolution as the cross-sections of the studied valleys are typically below this resolution. As ERA5 data is more realistic as it is a reanalysis product adjusted to observations and the aim of the training process was to detect foehn by its synoptic patterns, we decided to not alter the ERA5 data to be as accurate as possible during the training process. Downscaling with any common interpolation will not deliver any additional information or spatial patterns in the ERA5 fields, so increasing the resolution will not improve the training data. Mony et al. (2021) showed, that linking synoptic patterns to foehn is possible even with a resolution of around 80 km by using ERA-Interim.

2. Unclear how a foehn day is diagnosed from hourly measurements. The paper states "whereas the most extended event observed within the day's 24 hours is used". Does this mean that a "foehn day" is one with at least one hour of foehn?

You are correct, a day is considered as foehn day when at least one hour is detected satisfying OFC criteria. If a widespread and a localised foehn is located within one day, the widespread extent is used. This will be clarified.

3. Foehn diagnosis not reproducible and possibly inappropriate:

   a) Table 2 does not state a wind sector for foehn at a valley station. Please supply. If none was used but only the exceedance of 50 (90) % of wind speeds at the crest then e.g. situations with southerly flow aloft but diurnal upvalley winds will be misclassified.

   We agree that this might lead to misclassifications at individual valleys and cases, although the potential temperature criteria should mitigate some of them. Wind sectors for the valley stations were not used because we have no robust information which wind sectors correspond to foehn events in the location of every of the valley stations. Methods like Dürr (2008) rely on expert's judgement to decide the allowed wind corridor at the valley station and to account for channelling, as suggested by Drechsel and Mayr (2008), which would have been not feasible to do for every of the 21 stations. We tried to further mitigate this problem by applying stricter wind speed conditions at the crest as well as the valley to favour foehn, which has a high probability of breaking through into the valley.

   b) Wind speed thresholds at the crest station (given that the wind is in the appropriate sector) are unnecessary and eliminate many foehn hours. Potential temperature difference and appropriate wind at the valley station will suffice. No wonder that hardly any widespread events are classified if they require top 10 % wind speeds at crest.

   As addressed, the wind speed limits at the crest were chosen to account for our missing criteria of valley wind direction and to mitigate misclassification of other valley wind systems. The higher limit is used because we judged, that a pronounced wind speed at the crest is required to provide conditions for foehn to break through on a widespread scale. We therefore accepted the trade-off, that some weaker foehn hours are not classified as such, while misclassification of other valley winds is mitigated. Additionally, while we agree that some foehn

hours are eliminated, training is done on a daily basis and therefore, the hourly station data is aggregated to a whole day and as mentioned in 2), even one hour of satisfied OFC criteria suffices. This further ensures, that foehn events are detected, even if the onset and break-down are not timed correctly.

c) Missing results of the foehn diagnosis

We did not put enough emphasis on the validation of foehn diagnosis up to now. As stated in the response to the rejection, we strive to drastically improve this section of our manuscript.

4. In-sample model results: From what I read in the manuscript, the authors performed a hyperparameter grid search to determine the tuning parameters of the xgBoost model. The data used for the grid search can then no longer be used to determine the expected model performance.

As explained in the response to major comment 1 by referee 1, this was a mistake done by us and will be changed in the future version, as indeed we used a training and evaluation subset to train the models. Accuracy and other performance parameters will be given in terms of the evaluation data set.

**Minor Comments**

1. The title is too long and confusing as it implies that south foehn leads to climate change

The new title preposition is, in accordance with referee 1: "Analysing the future trends of synoptic patterns enabling south foehn over western Austria in CMIP5 EURO-CORDEX models"

2. Line 72: Does the statement of the "binary nature" of foehn contradict how it is treated in 3 categories in equ(1)?

The binary nature of foehn refers to the behaviour of foehn at one station, where either foehn can occur or not. The three classes on the other hand correspond to whole regions with multiple stations. The confusing statement will be edited.

3. Note that you do not need pressure data to use the foehn classification method of Vergeiner (2004). As this method checks whether air could have dry-adiabatically descended from the crest station to the valley station, reducing the temperature from the crest station to the valley station with a (linear) dry-adiabatic temperature gradient will do and only needs station elevations.

You are correct. But as we used potential temperature in all other locations, air pressure data was required to calculate it at the crest station Valluga, for which temperature data only is available. Therefore we decided to calculate it using the pressure from Galzig.

4. Table 2: Is the crest wind speed threshold chosen only from the wind speed distribution of winds from 135 - 225 degrees or from all directions (which would not be appropriate)?

The crest wind speed threshold is chosen from all available wind speed data, not just for the sector 135 - 225 degrees. As stated in the response to the major comments, the choice of strong wind speed criteria at the crest aims to minimise the misclassification of e.g. valley wind systems and therefore was chosen to be stricter, i.e. from all directions.

**References**

Ban, N., Caillaud, C., Coppola, E., Pichelli, E., Sobolowski, S., Adinolfi, M., Ahrens, B., Alias, A., Anders, I., Bastin, S., Belušić, D., Berthou, S., Brisson, E., Cardoso, R. M., Chan, S. C., Christensen, O. B., Fernández, J., Fita, L., Frisius, T., Gašparac, G., Giorgi, F., Goergen, K., Haugen, J. E., Hodnebrog, Ø., Kartsios, S., Katragkou, E., Kendon, E. J., Keuler, K., Lavin-Gullon, A., Lenderink, G., Leutwyler, D., Lorenz, T., Maraun, D., Mercogliano, P., Milovac, J., Panitz, H.-J., Raffa, M., Remedio, A. R., Schär, C., Soares, P. M. M., Srnec, L., Steensen, B. M., Stocchi, P., Tölle, M. H., Truhetz, H., Vergara-Temprado, J., de Vries, H., Warrach-Sagi, K., Wulfmeyer, V., and Zander, M. J.: The first multi-model ensemble of regional climate simulations at kilometer-scale resolution, part I: evaluation of precipitation, Climate Dynamics, 57, 275–302, https://doi.org/10.1007/s00382-021-05708-w, 2021.

Drechsel, S. and Mayr, G. J.: Objective Forecasting of Foehn Winds for a Subgrid-Scale Alpine Valley, Weather and Forecasting, 23, 205–218, https://doi.org/10.1175/2007WAF2006021.1, 2008.

Dürr, B.: Automatisiertes Verfahren zur Bestimmung von Föhn in Alpentälern, Tech. Rep. 223, Bundesamt für Meteorologie und Klimatologie, MeteoSchweiz, https://www.meteosuisse.admin.ch/dam/jcr:3ed2aec8-0901-417a-acc3-8be11cce440a/Foehnindex_Arbeitsbericht_223_Automatisiertes_Verfahren_zur_Bestimmung_von_Foehn_in_Alpentaelern_de.pdf, 2008.

Mony, C., Jansing, L., and Sprenger, M.: Evaluating Foehn Occurrence in a Changing Climate Based on Reanalysis and Climate Model Data Using Machine Learning, Weather and Forecasting, 36, 2039–2055, https://doi.org/10.1175/WAF-D-21-0036.1, 2021.

Plavcan, D., Mayr, G. J., and Zeileis, A.: Automatic and Probabilistic Foehn Diagnosis with a Statistical Mixture Model, Journal of Applied Meteorology and Climatology, 53, 652–659, https://doi.org/10.1175/JAMC-D-13-0267.1, 2014.

Vergeiner, J.: South föhn studies and a new föhn classification scheme in the Wipp and Inn Valley, PhD Thesis, Department of Meteorology and Geophysics, University of Innsbruck, https://www.researchgate.net/publication/228789957_South_fohn_studies_and_a_new_fohn_classification_scheme_in_the_Wipp_and_Inn_Valley/link/5def52314585159aa47112a8/download, 2004.

Zängl, G., Chimani, B., and Häberli, C.: Numerical Simulations of the Foehn in the Rhine Valley on 24 October 1999 (MAP IOP 10), Monthly Weather Review, 132, 368–389, https://doi.org/10.1175/1520-0493(2004)132<0368:NSOTFI>2.0.CO;2, 2004.